# Generative Models: What Do They Know? Do They Know Things? Let's Find Out!

## Abstract

Generative models excel at creating images that closely mimic real scenes, suggesting they inherently encode scene representations. We introduce INTRINSIC LoRA (I-LoRA), a general approach that uses Low-Rank Adaptation (LoRA) to discover scene intrinsics such as normals, depth, albedo, and shading from a wide array of generative models. I-LoRA is lightweight, adding minimally to the model's parameters and requiring very small datasets for this knowledge discovery. Our approach, applicable to Diffusion models, GANs, and Autoregressive models alike, generates intrinsics using the same output head as the original images. We show a correlation between the generative model's quality and the extracted intrinsics' accuracy through control experiments. Finally, scene intrinsics obtained by our method with just hundreds to thousands of labeled images, perform on par with those from supervised methods trained on millions of labeled examples.

## 1 Introduction

Generative models can produce high-quality images almost indistinguishable from real-world photographs. They seem to demonstrate a profound understanding of the world, capturing nuances of realistic object placement, appearance, and lighting conditions. Yet, it remains an open question how these models encode such detailed knowledge, and whether representations of scene intrinsics exist in these models and can be extracted explicitly.

**Why is this an important question?** Understanding the basis of generative models' realistic outputs could enhance our grasp of the physical world through computational eyes, potentially refining image generation and interpretation processes across various applications. Our primary motivation is to explore whether intrinsic properties can be extracted from various generative models using the same approach. This lays the groundwork for future applications, such as improving the primary generation capabilities of these models. Developing a universally optimal method for this extraction has been the ultimate goal, yet it has eluded researchers in both computer vision and other fields, such as NLP, where this effort has been ongoing for years (e.g., Belinkov & Glass (2019); Ettinger (2020)). Our work aims to fill this gap, providing valuable insights into the inner workings of generative image models and their potential applications.

Recent work has begun to study this question. Bhattad et al. (2023a) demonstrated that StyleGAN can encode important scene intrinsics. Similarly, Zhan et al. (2023) showed that diffusion models can understand 3D scenes in terms of geometry and shadows. Chen et al. (2023) found that Stable Diffusion's internal activations encode depth and saliency maps that can be extracted with linear probes. Three independent efforts (Luo et al., 2023b; Tang et al., 2023; Hedlin et al., 2023) discovered correspondences in diffusion models. However, these insights often pertain to specific models, leaving a gap in our understanding of whether such encoding is ubiquitous across generative architectures. Our work addresses this gap by introducing a uniform approach, INTRINSIC LoRA (I-LoRA) to extract intrinsic properties across multiple generative image model types, including GANs, Autoregressive, and Diffusion models.

**Why study all types of generative models?** Although diffusion models, such as Stable Diffusion (Rombach et al., 2022) and Imagen (Saharia et al., 2022), have garnered significant attention recently, the latest iterations of other model types like GigaGAN (Kang et al., 2023), CM3leon (Yu et al., 2023), and Parti (Yu et al., 2022)

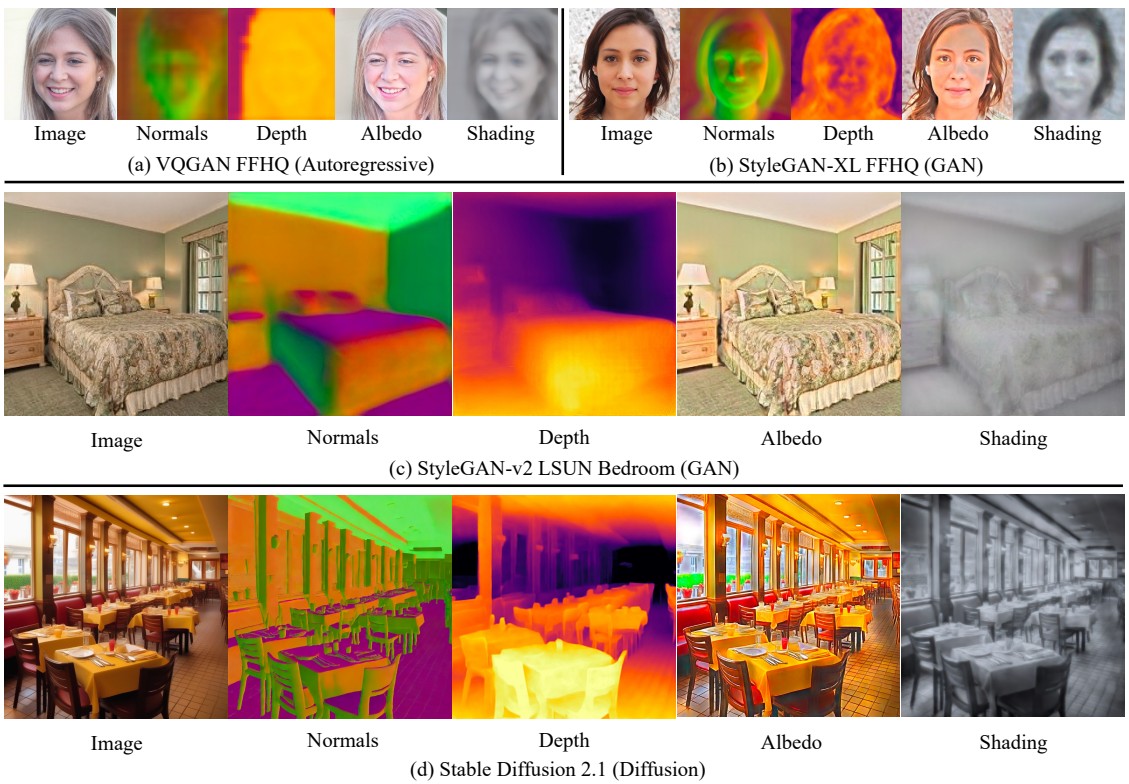

Figure 1: We propose INTRINSIC LoRA (I-LoRA), a model-agnostic, general approach for extracting visual knowledge from various generative models, including autoregressive, GANs, and diffusion models. Our method applies targeted, lightweight fine-tuning to modulate key feature maps, using low-rank adaptation (LoRA) on attention layers in VQGAN (a) and Stable Diffusion (d), and affine layers in StyleGAN (b and c). This allows us to discover fundamental scene intrinsics–normals, depth, albedo, and shading–directly from the models' learned representations, eliminating the need for additional task-specific decoding heads or layers.

have demonstrated a capacity to produce comparably high-quality images. Investigating these diverse models enables us to devise a general framework that applies not only to current models but is also adaptable to future developments and new generative models.

**Our Contributions and Findings.** We conduct our inquiry across a spectrum spanning diffusion, GANs, and autoregressive models to understand whether they encode fundamental scene intrinsics such as normals, depth, albedo, and shading (Barrow & Tenenbaum, 1978). We show that our method, I-LoRA, efficiently extracts these intrinsics across different model types with minimal computational overhead (less than 0.17% additional parameters for models like Stable Diffusion v1-5) and data requirements (as few as 250 labeled images). Importantly, I-LoRA does not require learning new heads, unlike Luo et al. (2023a) and Zhao et al. (2023), or complete fine-tuning (Zhao et al., 2023; Ke et al., 2023); instead, it predicts these intrinsic images from the same head or decoder that generates the images. This is significant because this simplified knowledge extraction reduces parameter requirements while maintaining efficiency and effectiveness.

A summary of our findings is presented in Tab. 1 and elaborated in Sec. 4. We observed significant improvement in the quality of recovered intrinsics when we used improved versions of Stable Diffusion (v1-1, v1-2, v1-5). In our control experiment, we found that randomly initialized Stable Diffusion architecture could not produce scene intrinsics. These findings suggest that the intrinsic knowledge within generative models is not accidental but a byproduct of large-scale learning to mimic image data. In summary, our contributions are:

- **Wide Applicability:** We validate I-LoRA 's capability to extract scene intrinsics (normals, depth, albedo and shading) across a broad spectrum of generative models, highlighting its adaptability to diverse architectures. We also show our method can be extended to non-generative models.

Table 1: Summary of scene intrinsics found across different generative models without changing generator head. ✓: Intrinsics can be extracted with high quality. ∼: Intrinsics cannot be extracted with high quality. ✗: Intrinsics cannot be extracted.

| Model | Pretrain Type | Domain | Normal | Depth | Albedo | Shading |
|---|---|---|---|---|---|---|
| VQGAN (Esser et al., 2020) | Autoregressive | FFHQ | ∼ | ∼ | ✓ | ✓ |
| SG-v2 (Karras et al., 2020b) | GAN | FFHQ | ✓ | ∼ | ✓ | ✓ |
| SG-v2 (Yu et al., 2021) | GAN | LSUN Bed | ✓ | ✓ | ✓ | ✓ |
| SG-XL (Sauer et al., 2022) | GAN | FFHQ | ✓ | ∼ | ✓ | ✓ |
| SG-XL (Sauer et al., 2022) | GAN | ImageNet | ✗ | ✗ | ✗ | ✗ |
| SD-UNet (single-step) (Rombach et al., 2022) | Diffusion | Open | ✓ | ✓ | ✓ | ✓ |
| SD (multi-step) (Rombach et al., 2022) | Diffusion | Open | ✓ | ✓ | ✓ | ✓ |

- **Efficient and Lean Approach** to knowledge extraction: I-LoRA is highly efficient, requiring a little increase in parameters (less than 0.17% or as few as 0.04% for Stable Diffusion) and minimal training data, as few as 250 images.

- **Insights from Learned Priors:** Through control experiments with various Stable Diffusion configurations, we illustrate the role of learned priors in facilitating intrinsic representation, suggesting that the quality of intrinsics extracted is correlated to the visual quality of the generative model.

- **Competitive Quality of Intrinsics:** Our method, supervised with hundreds to thousands of labeled images, generates intrinsics on par with or even better than those produced by the leading supervised techniques requiring millions of labeled images. Moreover, our method outperforms linear probing and full finetuning of models in the limited data regime scenarios.

## 2 Related Work

**Generative Models:** Generative Adversarial Networks (GANs) (Goodfellow et al., 2014) have been widely used for generating realistic images. Variants like StyleGAN (Karras et al., 2019), StyleGAN2 (Karras et al., 2020b) and GigaGAN (Kang et al., 2023) have pushed the boundaries in terms of image quality and control. Some work has explored the interpretability of GANs (Bau et al., 2020; Bhattad et al., 2023a), but little is known about their ability to capture scene intrinsics.

Diffusion models, such as Denoising Score Matching (Vincent, 2011) and Noise-Contrastive Estimation (Gutmann & Hyvärinen, 2010), have been used for generative tasks and are perhaps the most popular at the moment (Karras et al., 2022; Ho et al., 2020; Rombach et al., 2022). These models have been shown to understand complex scene intrinsics like geometry and shadows (Zhan et al., 2023), but the generalizability of this understanding across different scene intrinsics is largely unexplored.

Autoregressive models like PixelRNN (Van Den Oord et al., 2016) and PixelCNN (Van den Oord et al., 2016) generate images pixel-by-pixel, offering fine-grained control but at the cost of computational efficiency. More recently, VQ-VAE-2 (Razavi et al., 2019) and VQGAN (Esser et al., 2020) have combined autoregressive models with vector quantization to achieve high-quality image synthesis. While these models are powerful, their ability to capture and represent scene intrinsics has not been thoroughly investigated.

**Scene Intrinsics Extraction:** Barrow & Tenenbaum (1978) highlighted several fundamental scene intrinsics including depth, albedo, shading, and surface normals. A large body of work has focused on extracting some related properties, like depth and normals from images (Eigen et al., 2014; Long et al., 2015; Eftekhar et al., 2021; Kar et al., 2022; Ranftl et al., 2021; Bhat et al., 2023) using labeled annotated data. Labeled annotations of albedo and shading are hard to find and as the recent review in Forsyth & Rock (2021) shows, methods involving little or no learning have remained competitive. However, these methods often rely on supervised learning and do not explore the capabilities of generative models in this context.

Many recent studies have used generative models as pre-trained feature extractors or scene prior learners. They use generated images to enhance downstream discriminative models, fine-tune the original generative

model for a new task, learn new layers or decoders to produce desired scene intrinsics (Abdal et al., 2021; Jahanian et al., 2021; Zhang et al., 2021b; Li et al., 2021; Noguchi & Harada, 2020; Bao et al., 2022; Xu et al., 2023; Sariyildiz et al., 2023; Zhao et al., 2023; Ke et al., 2023). InstructCV (Gan et al., 2023) executes computer vision tasks via natural language instructions, abstracting task-specific design choices. However, it requires re-training of the entire diffusion model. In contrast, we show that many generative models capture intrinsic image knowledge implicitly and do not require specialized training to extract this information.

**Knowledge in Generative Models:** Several studies have explored the extent of StyleGAN's knowledge, particularly in the context of 3D information about faces (Pan et al., 2021; Zhang et al., 2021a). Yang et al. (2021) show GANs encode hierarchical semantic information across different layers. Further research has demonstrated that manipulating offsets in StyleGAN can lead to effective relighting of images (Bhattad et al., 2024; 2023b) and extraction of scene intrinsics (Bhattad et al., 2023a). Chen et al. (2023) found internal activations of the LDM encode linear representations of both depth data and a salient-object / background distinction. Wu et al. (2023) also demonstrate rich latent codes of diffusion models can be easily mapped to annotations with small amount of training samples. Recently, Tang et al. (2023); Luo et al. (2023b); Hedlin et al. (2023) found correspondence emerges in image diffusion models. Sarkar et al. (2023) showed generative models struggle to replicate accurate projective geometry in generated images.

Concurrent studies, such as the one by Luo et al. (2023a), have explored training task-specific "readout" networks to extract signals like pose, depth, and edges from feature maps in Stable Diffusion models for controlling image generation. Our goals are different: We are interested in recovering intrinsic images, while the aim of Luo et al. (2023a) is controlling image generation. Our method offers notable advantages in parameter efficiency: Our I-LoRA is approximately 5 times more parameter-efficient than readout networks in their application to SD v1-5 (compare 8.5M vs 1.59M). Lastly, the broad applicability of "readout" networks across various generative model types remains uncertain.

Another concurrent work Lee et al. (2023) applies a LoRA-like approach to adapt a pre-trained diffusion model for dense semantic tasks. Our work differs from theirs in several aspects: First, their goal is to use pre-trained diffusion models as strong priors for dense prediction. Second, their tasks are within restricted domains, such as bedrooms. Finally, they do not extend to the wide range of generative models our study explores. Our paper not only demonstrates I-LoRA's efficacy across different model architectures but also explores its application in a diverse array of scene contexts including real images.

**Relation to Fine-Tuning and Linear Probing.** Previous approaches for depth extraction from generative models fine-tune the entire model (Zhao et al., 2023; Ke et al., 2023) or apply linear probing (Chen et al., 2023). Fine-tuning alters the entire model with a specific dataset, transforming it into a new version that might lose its original image-generating capabilities. This raises questions about whether the depth perception was an inherent quality of the model or a result of the fine-tuning process. A drawback of linear probing lies in modeling each layer independently. While the tiny parameters, samples, and iterations required to train I-LoRA indicate that intrinsic information is easy to surface in generative models, the weak performance of linear probes indicates this knowledge is distributed throughout the network.

In contrast, I-LoRA performs efficient fine-tuning, and leverages the entire network's learned representation, modulating the internal flow of information for intrinsic extraction. This approach not only ensures a more holistic utilization of the network but also proves to be superior, as evidenced by our ablation studies. Another important property of I-LoRA is its preservation of the original's model output head to extract scene intrinsics—normals, depth, albedo, and shading as 3-channel, image-like maps, without requiring any new decoding layers. By doing so, it retains the original image generation functionality, while accessing the latent visual knowledge within generative models.

**LoRA (Low-Rank Adaptation).** LoRA (Hu et al., 2022), originally proposed to reduce the cost of fine-tuning large language models for downstream tasks, introduces trainable low-rank decomposed matrices into specific layers of the model architecture. These matrices are the only components updated during task-specific optimization. This results in a significant reduction in the number of trainable parameters, ensuring only slight modifications to the model, and preserving its core functionality and accessibility.

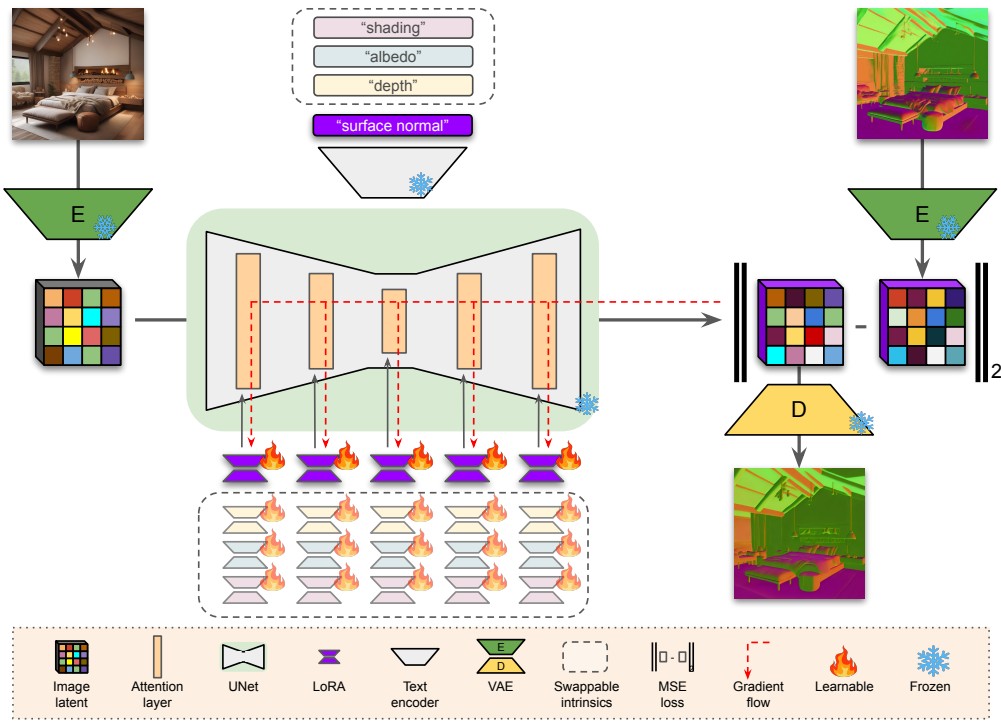

Figure 2: Overview of I-LoRA applied to Stable Diffusion's UNet in a single-step manner. We adopt an efficient fine-tuning approach, specifically low-rank matrices corresponding to key feature maps – attention matrices – to reveal scene intrinsics. Distinct low-rank adaptors (LoRA) are optimized for each intrinsic (**_violet_** adaptors for surface normals; swappable with other intrinsics). We use a few labeled examples for this fine-tuning and directly extract scene intrinsics using the same decoder that generates images, circumventing the need for specialized decoders or comprehensive model re-training.

## 3   Intrinsic LoRA

A generative model $G$ maps noise/conditioning information $z$ to an RGB image $G(z) \in \mathbb{R}^{H \times W \times 3}$. We seek to augment $G$ with a small set of parameters $\theta$ that allow us to produce, using the same architecture as $G$, an image-like map with up to three channels, representing scene intrinsics like surface normals.

**I-LoRA's Learning Framework.** Our method, INTRINSIC LoRA(I-LoRA), learns to extract intrinsic properties of an image (such as depth) using a small number of labeled examples (image/depth map pairs) as supervision. In cases where we do not have access to the actual intrinsic properties, we use models trained on large datasets to generate estimated intrinsics (such as estimated depth for an image) as pseudo-ground truth, used as training targets for $G_\theta$. To optimize $\theta$ of $G_\theta$ using a pseudo-ground truth predictor $\Phi$, we minimize the objective:

$$\min_\theta \mathbb{E}_z[d(G_\theta(z), \Phi(G(z)))], \tag{1}$$

where $d$ is a distance metric that depends on the intrinsics we wish to learn.

Diffusion models require special treatment since they are effectively image-to-image and not noise-to-image. During inference, diffusion models repeatedly receive a noisy image as input. Thus instead of conditioning noise $z$ we feed an image $x$(generated or real) to a diffusion model $G$. In this case, given a real image $x$, our objective function becomes $\min_\theta \mathbb{E}_x[d(G_\theta(x), \Phi(x))]$.

For surface normals $\Phi$ is Omnidatav2-Normal (Eftekhar et al., 2021; Kar et al., 2022). To generate pseudo ground truth for depth we use ZoeDepth (Bhat et al., 2023) as the predictor $\Phi$. For Albedo and Shading $\Phi$ is Paradigms (Forsyth & Rock, 2021; Bhattad & Forsyth, 2022). For SG2, SGXL and VQGAN, $d$ in Eq.1 is

$$d(x, y) = 1 - cos(x, y) + \|x - y\|_1 \tag{2}$$

for normal and MSE for other intrinsics. For latent diffusion, there isn't a clear physical meaning to the relative angle of latent vectors in encoded normals, so we use the standard objective of MSE for all intrinsics.

We use LoRA, a parameter-efficient adaptation technique, to recover image intrinsics from generative models. LoRA introduces a low-rank weight matrix $W^*$, which has a lower rank than the original weight matrix $W \in \mathbb{R}^{d_1 \times d_2}$. This is achieved by factorizing $W^*$ into two smaller matrices $W_u^* \in \mathbb{R}^{d_1 \times d^*}$ and $W_l^* \in \mathbb{R}^{d^* \times d_2}$, where $d^*$ is chosen such that $d^* \ll \min(d_1, d_2)$. The output $o$ for an input activation $a$ is then given by:

$$o = Wa + W^*a = Wa + W_u^*W_l^*a. \tag{3}$$

To preserve the original model's behavior at initialization, $W_u^*$ is initialized to zero. We next describe how we leverage LoRA modules to extract intrinsics from Diffusion models, GANs, and Autoregressive models.

**Applying I-LoRA.** For **diffusion models**, I-LoRA adaptors are learned atop cross-attention and self-attention layers. The UNet is utilized as a dense predictor, transforming an RGB input into intrinsics in one step. This approach, favoring simplicity and effectiveness, delivers superior quantitative results. Depending on the intrinsic of interest, the textual input varies among "surface normal", "depth", "albedo", or "shading". Fig. 2 illustrates the I-LoRA pipeline. For **GANs**, I-LoRA modules are integrated with the affine layers that map from w-space to s-space (Wu et al., 2021). In the case of **VQGAN, an autoregressive model**, I-LoRA is applied to the convolutional attention layers within the decoder.

## 4 Experiments

In this section, we outline I-LoRA's contributions, demonstrating its general applicability across generative models (Sec. 4.1) and its efficiency in terms of parameters and labeling (Sec. 4.2). Control experiments provide evidence of I-LoRA's effectiveness (Sec. 4.3), while comparative analysis establish its superiority over both fine-tuning and linear probing methods (Sec. 4.4). Additional ablation studies and baseline comparisons further confirm I-LoRA's robustness (Appendix B). Note: our analysis in Sec. 4.2 uses a single-step I-LoRA model for intrinsic image extraction from stable diffusion. In Sec. 5, we discuss the challenge of naively applying I-LoRA to a multi-step Stable Diffusion model. To address this, we propose a simple modification to the architecture by adding an extra, non-learned layer for improved intrinsic image extraction. We refer to this modified model as **Augmented I-LoRA** (I-LoRA_AUG).

### 4.1 I-LoRA is General and Universally Applicable

We evaluate I-LoRA across diverse generative models, including StyleGAN-v2 (Yu & Smith, 2019), StyleGAN-XL (Sauer et al., 2022), and VQGAN (Esser et al., 2020), trained on datasets like FFHQ (Karras et al., 2020b), LSUN Bedrooms (Yu et al., 2015), and ImageNet (Deng et al., 2009). I-LoRA adaptors are tailored to each model and dataset to extract intrinsics: surface normals, depth, albedo, and shading, demonstrating broad applicability and robustness in both qualitative assessments (Fig. 1, 3, 4, 6) and quantitative (Tab. 2 on generated images, Tab. 3 on real images). In all experiments – covering both generated and real images – we use pseudo-ground truth from pre-trained models as a supervisory signal for fine-tuning I-LoRA to discover scene intrinsics within generative models as previously mentioned in Sec. 3. We use I-LoRA with Rank 8 as default for all generative models if not otherwise mentioned.

We find I-LoRA can unearth intrinsic knowledge across almost all models tested, the notable exception is StyleGAN-XL trained on ImageNet. Where it yields qualitatively poor results, which we attribute to the model's limited ability to generate realistic images (Fig. 5). This suggests the quality of intrinsic extraction is correlated with the generative model's fidelity (see Sec. 4.3). In evaluations of generated images, our method is benchmarked against pseudo-ground truths derived from existing models, compensating for the lack of true ground truths. The performance of I-LoRA , gauged through these comparisons, provides useful indicators but must be interpreted within the context of the selected pseudo-ground truths.

Thanks to their architecture as image-to-image translators, diffusion models excel as powerful image generators. This feature simplifies their application to real images. Taking advantage of this, we apply I-LoRA to directly extract intrinsic images from Stable Diffusion's UNet in a single step. This method bypasses the

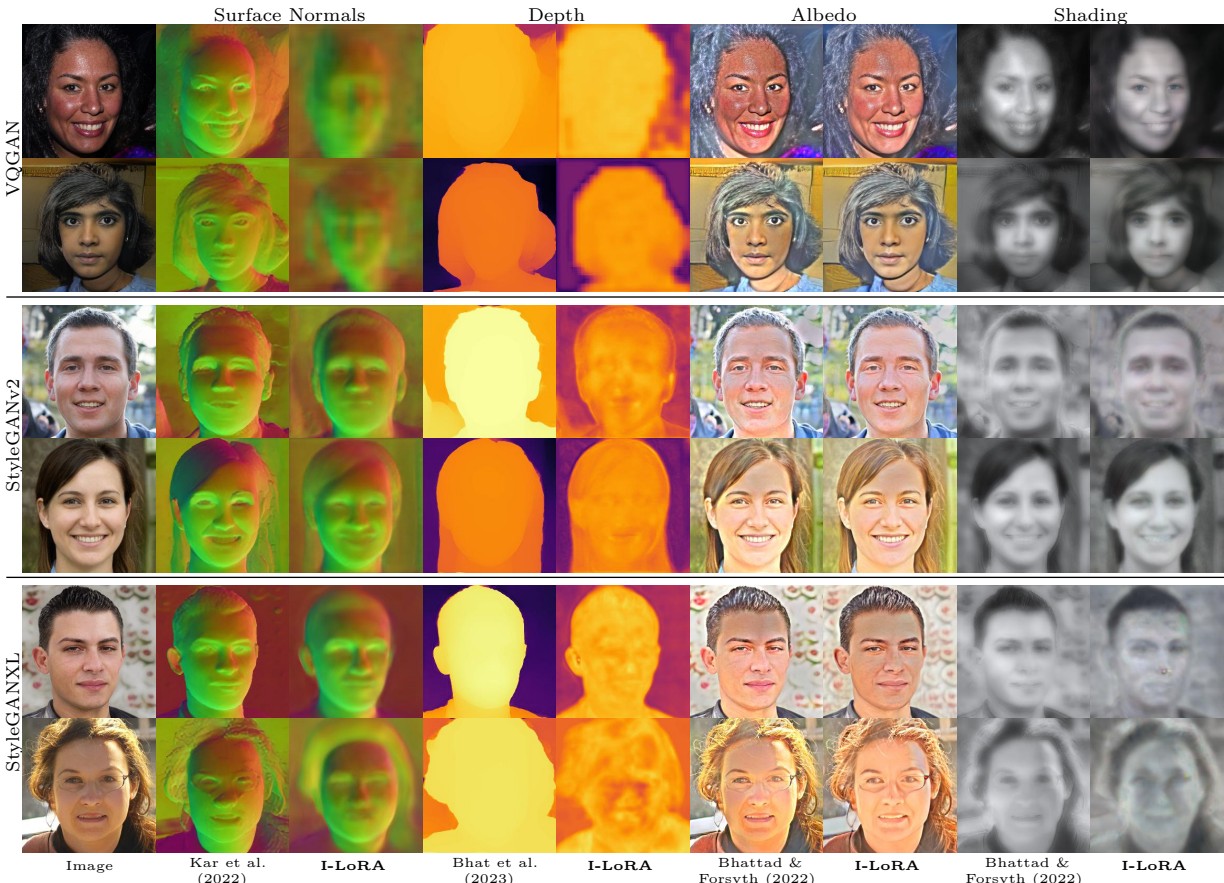

Figure 3: Scene intrinsics from different generators – VQGAN, StyleGAN-v2, and StyleGAN-XL – trained on FFHQ dataset: The "image" column shows the synthetic images produced by each model. Subsequent columns show four scene intrinsics extracted by a SOTA non-generative model and I-LoRA (ours).

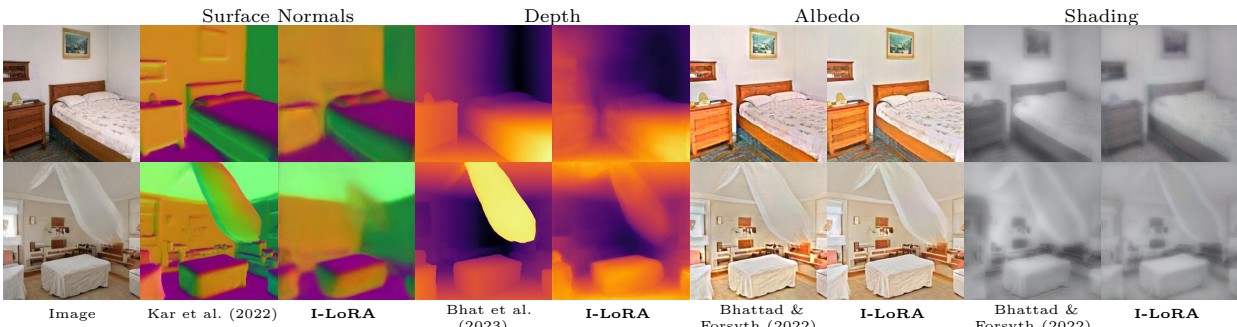

Figure 4: I-LoRA's extracted scene intrinsics from StyleGAN-v2 trained on LSUN bedroom images are competitive with task-specific models.

iterative reverse denoising process. The model receives a real image as input and outputs the corresponding image intrinsics through I-LoRA. Such direct application allows for evaluation against actual ground truth. This provides a definitive benchmark for assessing I-LoRA's effectiveness on DIODE dataset (Vasiljevic et al., 2019). We use the official training/evaluation split in all of our DIODE experiments. For training with fewer samples, we randomly chose samples from the official training partition. All the metrics we reported on DIODE are computed over the entire evaluation set. In Tab. 3, we find that I-LoRA not only matches but, in several metrics (median error for surface normals, RMSE for depth), surpasses the performance of Omnidata and ZoeDepth – the source of its training signal – while using significantly less data, parameters, and training time (see Sec.4.2).

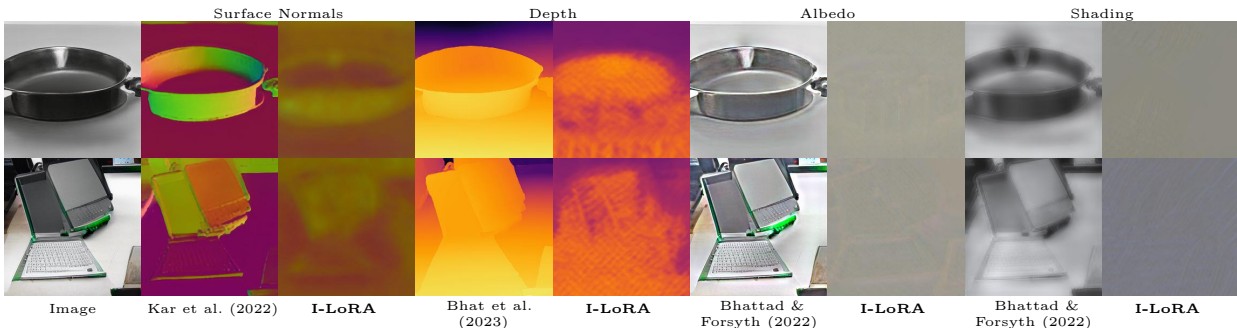

Figure 5: StyleGAN-XL trained on ImageNet. Top: pan, bottom: laptop, with the corresponding scene intrinsics (pseudo ground truth and extracted) alongside. The surface normals and depth maps, while capturing the basic shape and volume, lack precise detail and exhibit artifacts. Albedo and Shading extractions fail. These difficulties are correlated with the overall worse realism and consistency of the generated images.

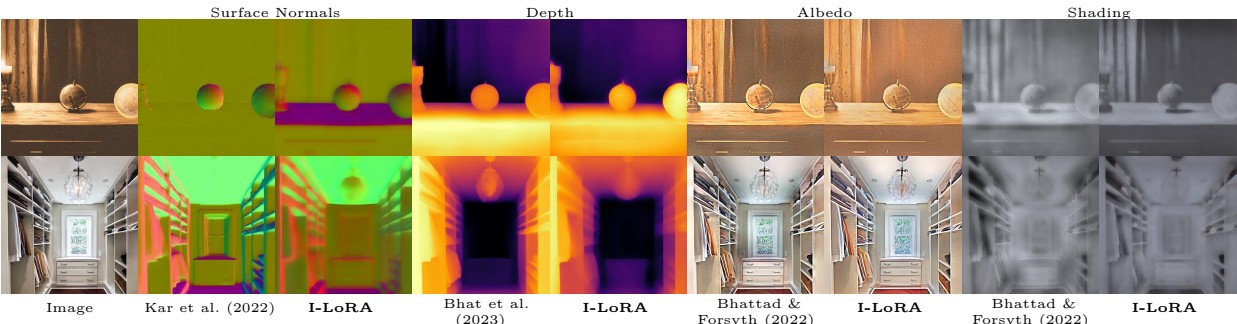

Figure 6: Scene intrinsics from I-LoRA applied to randomly generated images. I-LoRA accurately predicts the table's normal in the first row when compared to Kar et al. (2022). The globe in the right corner also appears to be closer to the camera in our depth compared to Bhat et al. (2023). In the second row, ceiling lamp normals are also visible in I-LoRA but not in Kar et al. (2022). The comparison highlights I-LoRA's ability to closely align with, and sometimes surpass, these supervised SOTA monocular predictors.

Table 2: Quantiative analysis of scene intrinsics extraction performance by I-LoRA on generated images. We compare with pseudo ground truths from Omnidata-v2 for surface normals, ZoeDepth for depth, and Paradigms for albedo and shading. Metrics include mean angular error, median angular error, and L1 error for surface normals; RMS and $\delta < 1.25$ for depth; RMS for albedo and shading.

| Model | Pre-training Type | Domain | LoRA Param. | Surface Normal | | | Depth | | Albedo | Shading |
|---|---|---|---|---|---|---|---|---|---|---|
| | | | | Mean Error°↓ | Median Error°↓ | L1 Error$_{\times 100}$ ↓ | RMS ↓ | $\delta < 1.25_{\times 100\%}$ ↑ | RMS ↓ | RMS ↓ |
| VQGAN | Autoregressive | FFHQ | 0.18% | 19.97 | 20.97 | 16.33 | 0.1819 | 62.33 | 0.0345 | 0.0106 |
| StyleGAN-v2 | GAN | FFHQ | 0.57% | 16.93 | 19.60 | 13.87 | 0.1530 | 90.74 | 0.0283 | 0.0110 |
| StyleGAN-XL | GAN | FFHQ | 0.29% | 15.28 | 18.07 | 12.63 | 0.1337 | 93.87 | 0.0287 | 0.0125 |
| StyleGAN-v2 | GAN | LSUN Bedroom | 0.57% | 13.94 | 24.76 | 11.49 | 0.0897 | 66.88 | 0.0270 | 0.0074 |
| StyleGAN-XL | GAN | ImageNet | 0.29% | 24.09 | 25.52 | 19.44 | 0.2175 | 38.38 | 0.1065 | 0.0119 |
| I-LoRA$_{\text{AUG}}$ (multi step) | Diffusion | Open | 0.17% | 21.41 | 28.57 | 17.39 | 0.2042 | 41.21 | 0.0881 | 0.0099 |
| I-LoRA (single step) | Diffusion | Open | 0.17% | 16.63 | 23.64 | 13.69 | 0.1179 | 52.59 | 0.0487 | 0.0118 |

**Extending I-LoRA to DINO.** I-LoRA can extend beyond generative models to include self-supervised, non-generative models like DINO (Darcet et al., 2023). To explore this possibility, albeit tangential to our main objective of extracting intrinsic knowledge from generative models, we follow Oquab et al. (2023) by learning a linear head to project DINO features to pixel space, along with our I-LoRA modules. Using DINOv2's "giant" model, we achieve quantitative results on par with our I-LoRA at the cost of only 0.26% extra parameters. However, qualitatively, DINOv2 produces less smooth intrinsics (Fig. 7d) with apparent discontinuities. This broad applicability echoes Bhattad et al. (2023a)'s "meaningful image representations are those that can capture these intrinsic properties", a hypothesis validated across models of various types.

Table 3: Quantitative analysis of scene intrinsic extraction performance across different models on real images.

| Model | Pre-training Type | LoRA Param | Surface Normal | | | Depth | |
|---|---|---|---|---|---|---|---|
| | | | Mean Error°↓ | Median Error°↓ | L1 Error$_{\times 100}$ ↓ | RMS ↓ | $\delta < 1.25_{\times 100}$ ↑ |
| Omnidata-v2 (Kar et al., 2022)/ZoeDepth (Bhat et al., 2023) | Supervised | - | **18.90** | 13.36 | **15.21** | 0.2693 | **47.56** |
| DINOv2 | Non-Generative | 0.26% | 19.74 | 13.72 | 16.00 | 0.2094 | 44.32 |
| I-LoRA$_{AUG}$ (multi step) | Diffusion | 0.17% | 23.74 | 19.08 | 19.31 | 0.2651 | 43.19 |
| I-LoRA (single step) | Diffusion | 0.17% | 20.31 | **12.54** | 16.53 | **0.2046** | 44.90 |

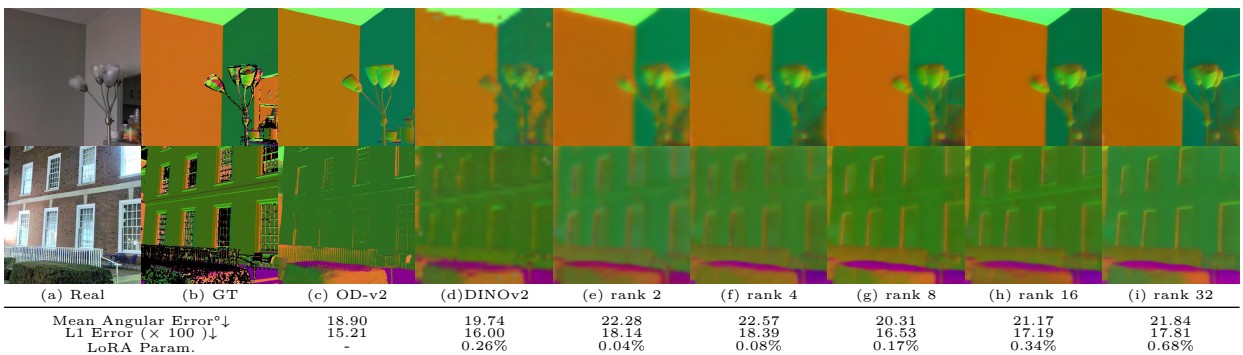

| | (a) Real | (b) GT | (c) OD-v2 | (d)DINOv2 | (e) rank 2 | (f) rank 4 | (g) rank 8 | (h) rank 16 | (i) rank 32 |
|---|---|---|---|---|---|---|---|---|---|
| Mean Angular Error°↓ | | 18.90 | 19.74 | 22.28 | 22.57 | 20.31 | 21.17 | 21.84 |
| L1 Error (× 100 )↓ | | 15.21 | 16.00 | 18.14 | 18.39 | 16.53 | 17.19 | 17.81 |
| LoRA Param. | | - | 0.26% | 0.04% | 0.08% | 0.17% | 0.34% | 0.68% |

Figure 7: Parameter Efficiency of I-LoRA. We evaluate I-LoRA across various rank settings for surface normal extraction. Lower ranks such as 8 offer a balance between efficiency and effectiveness. All model variants are trained using SD's UNet (v1.5) with 4000 samples. Performance metrics, such as Mean Angular Error and L1 Error for normals, and additional parameter counts are detailed below each variant.

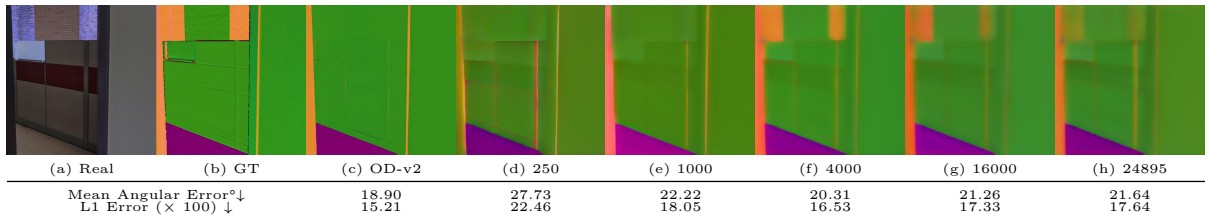

| | (a) Real | (b) GT | (c) OD-v2 | (d) 250 | (e) 1000 | (f) 4000 | (g) 16000 | (h) 24895 |
|---|---|---|---|---|---|---|---|---|
| Mean Angular Error°↓ | | 18.90 | 27.73 | 22.22 | 20.31 | 21.26 | 21.64 |
| L1 Error (× 100) ↓ | | 15.21 | 22.46 | 18.05 | 16.53 | 17.33 | 17.64 |

Figure 8: Data efficiency of I-LoRA. Note: SOTA supervised model (c), was trained using 12M+ labeled training samples. Even with 250 samples, I-LoRA captures surface normals. We observe the best performance with 4k samples. Models (d)-(h) all use the same SD UNet(v1-5) and rank 8 LoRA.

## 4.2 I-LoRA is Parameter and Label Efficient

Our single-step I-LoRA model, distinguished by its high quantitative performance, serves as the basis for ablation studies that assess the influence of rank and labeled data quantity on intrinsic extraction efficiency. We verify that the requirements for compute, parameters, and data to learn I-LoRA are minimal.

**Parameter efficiency.** Fig. 7 shows surface normal predictions across LoRA ranks. The highest accuracy is achieved with Rank 8, balancing accuracy and memory. Notably, a Rank 2 LoRA with only 0.4M additional parameters (a mere 0.04% increase) still yields good performance. Note that across different generative models, Rank 8 adaptors adds only 0.17% to 0.57% additional parameters (Tab. 2).
**Label efficiency.** The impact of the labeled data size is analyzed in Fig. 8. I-LoRA reaches peak performance using a modest 4000 training examples, with credible predictions visible from as few as 250 samples.

## 4.3 Control Experiments and Correlation with Generative Quality

To assess if our I-LoRA leverages pre-trained generative capabilities or primarily depends on LoRA layers, we performed a control experiment using a randomly initialized SD UNet, following the same training protocol of our I-LoRA model. The poor results from this model (see Fig. 9) corroborate that the learned features developed during generative pre-training are crucial for intrinsic extraction, rather than I-LoRA layers

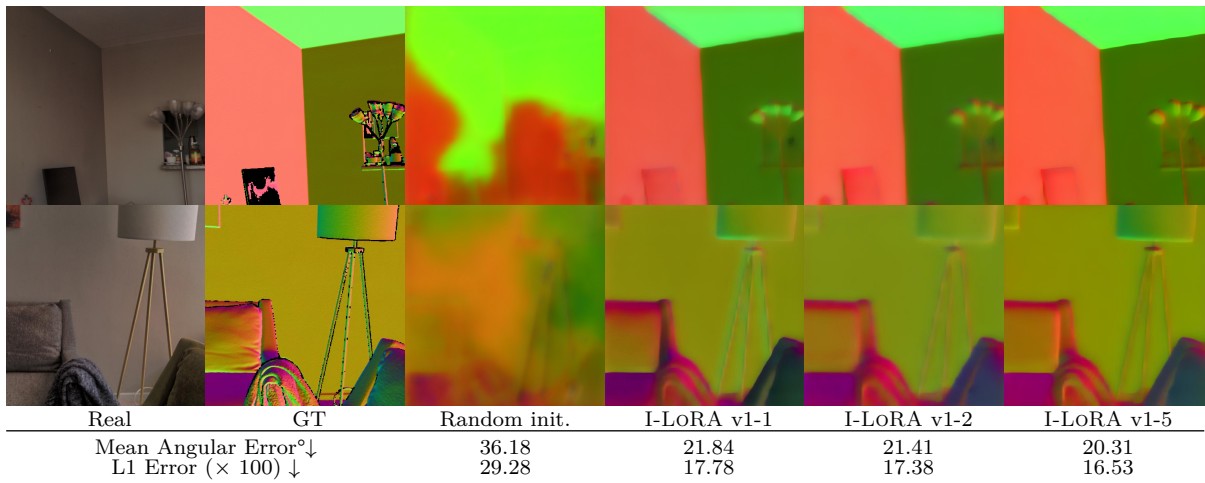

| | Real | GT | Random init. | I-LoRA v1-1 | I-LoRA v1-2 | I-LoRA v1-5 |
|---|---|---|---|---|---|---|
| Mean Angular Error°↓ | | | 36.18 | 21.84 | 21.41 | 20.31 |
| L1 Error (× 100) ↓ | | | 29.28 | 17.78 | 17.38 | 16.53 |

Figure 9: We find a correlation between generative model quality and scene intrinsic extraction accuracy. We compare different versions of Stable Diffusion (v1-1, v1-2, v1-5). The progress from SD v1-1 to SD v1-5 shows improvements in intrinsic extraction paralleling improvements in image generation. Control experiments with a randomly initialized UNet fail to extract surface normals, emphasizing the reliance on learned priors from generative training for effective intrinsic representation extraction.

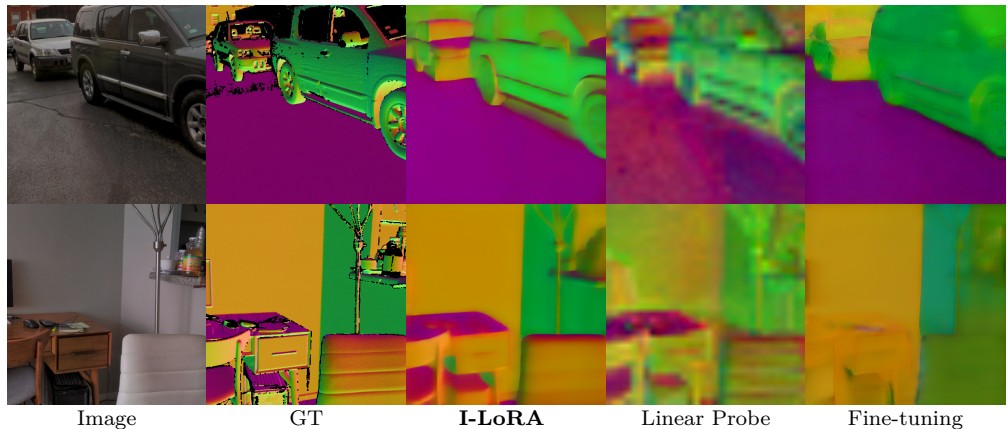

| Image | GT | **I-LoRA** | Linear Probe | Fine-tuning |
|---|---|---|---|---|

Figure 10: Comparison with baselines. All models are trained with 250 samples. Note LoRA effectively extracts better normals compared to other baselines.

Table 4: We find I-LoRA to consistently outperform baselines for different training samples (first row).

| | Steps/s | Peak Train GPU Mem% | 250 | | 1000 | | 4000 | | 16000 | |
|---|---|---|---|---|---|---|---|---|---|---|
| | | | Mean Error°↓ | L1 $_{\times 100}$ ↓ | Mean Error°↓ | L1 $_{\times 100}$ ↓ | Mean Error°↓ | L1 $_{\times 100}$ ↓ | Mean Error°↓ | L1 $_{\times 100}$ ↓ |
| Linear Probe | 2.13 | 29.46% | 29.10 | 23.74 | 28.45 | 23.25 | 28.52 | 23.26 | 28.22 | 23.11 |
| Fine-tuning | 0.77 | 86.78% | 34.40 | 27.58 | 25.19 | 20.28 | 28.03 | 22.17 | 27.39 | 22.24 |
| LoRA (Ours) | 0.94 | 63.48% | **27.73** | **22.46** | **22.22** | **18.05** | **20.31** | **16.53** | **21.26** | **17.33** |

alone. Furthermore, analyzing multiple Stable Diffusion versions (v1-1, v1-2 and v1-5) under the same I-LoRA protocol reveals that enhancements in image generation quality correlate positively with intrinsic extraction capabilities. This assertion is further reinforced by observing a correlation between lower FID scores (9.6 for VQGAN (Esser et al., 2020), 3.62 for StyleGAN-v2 (Karras et al., 2020a) and 2.19 for StyleGAN-XL (Sauer et al., 2022)) and improved intrinsic predictions in our FFHQ experiments (Fig. 3 and Tab. 2: first three rows), confirming that superior generative models yield more accurate intrinsics.

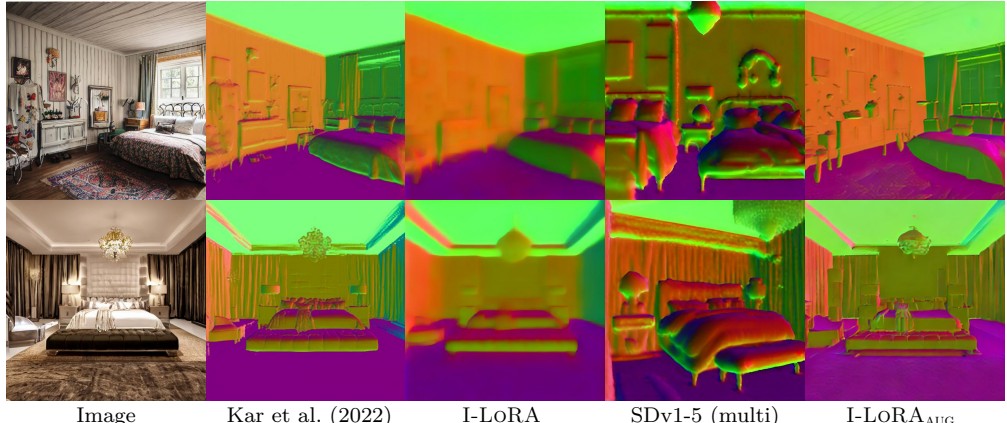

| Image | Kar et al. (2022) | I-LoRA | SDv1-5 (multi) | I-LoRA$_{\text{AUG}}$ |

Figure 11: I-LoRA yields satisfactory results, but multiple diffusion steps lead to misalignment in extracted intrinsics (fourth column). The last column, I-LoRA$_{\text{AUG}}$ , demonstrates successfully correcting the misalignment using our image conditioning approach, resulting in well-aligned and detailed intrinsic extractions

### 4.4 Superiority of I-LoRA over Fine-tuning and Linear Probing

We compare I-LoRA with two common baselines: linear probing and full model fine-tuning. Following Chen et al. (2023) for linear probing and employing standard fine-tuning practices, we train all methods with a small dataset of 250 samples to 16000 samples. All three models are trained with the same number of epochs and have converged at the end of the training. Our findings in Tab. 4 and Fig. 10 show that I-LoRA significantly outperforms these baselines in low-data regimes, validating its superior efficacy and data efficiency.

## 5 I-LoRA$_{\text{aug}}$ : Towards Improved Intrinsic Extraction

In the previous section, we showed that Stable Diffusion models inherently capture various scene intrinsics like normals, depth, albedo, and shading, as evidenced by our evaluation of I-LoRA . A natural question arises: can we enhance these intrinsics using multi-step diffusion inference? While multi-step diffusion improves sharpness, we find it introduces two challenges: 1. intrinsics misaligned with input, and 2. shift in the distribution of outputs relative to the ground truth (visually manifesting as a color shift) (see Fig. 11).

To address the first challenge, we augment the noise input to the UNet with the input image's latent encoding, as in InstructPix2Pix (Brooks et al., 2023). The second challenge is a known artifact attributed to Stable Diffusion's difficulty generating images that are not with medium brightness (Deck & Bischoff, 2023; Lin et al., 2023). Lin et al. (2023) propose a Zero SNR strategy that reduces color discrepancies but requires diffusion models trained with v-prediction objective, which SDv1-5 does not. However, Stable Diffusion v2-1 employs a v-prediction objective. Therefore we replace SDv1-5 with SDv2-1 while maintaining our previously described learning protocol. We name this multi-step augmented SDv2-1 model I-LoRA$_{\text{AUG}}$. I-LoRA$_{\text{AUG}}$ solves the misalignment issue and reduces the color shift significantly (Fig. 12), resulting in the generation of high-quality, sharp scene intrinsics with improved quantitative accuracy. However, quantitatively, the results still fall short of our single-step I-LoRA result. In the future, we hope this problem will be solved by improved sampling techniques and the next generation of generative models.

## 6 Discussions, Limitations and Broader Impact

We find consistent evidence that generative models implicitly learn physical scene intrinsics, allowing tiny LoRA adaptors to extract this information with minimal fine-tuning on small labeled data. More powerful generative models produce more accurate scene intrinsics, strengthening our hypothesis that learning this information is a natural byproduct of learning to generate images well. Additionally, we discovered scene intrinsics exist across generative models and the self-supervised DINOv2, resonating with Barrow & Tenenbaum (1978)'s hypothesis of fundamental "scene characteristics" emerging in visual processing.

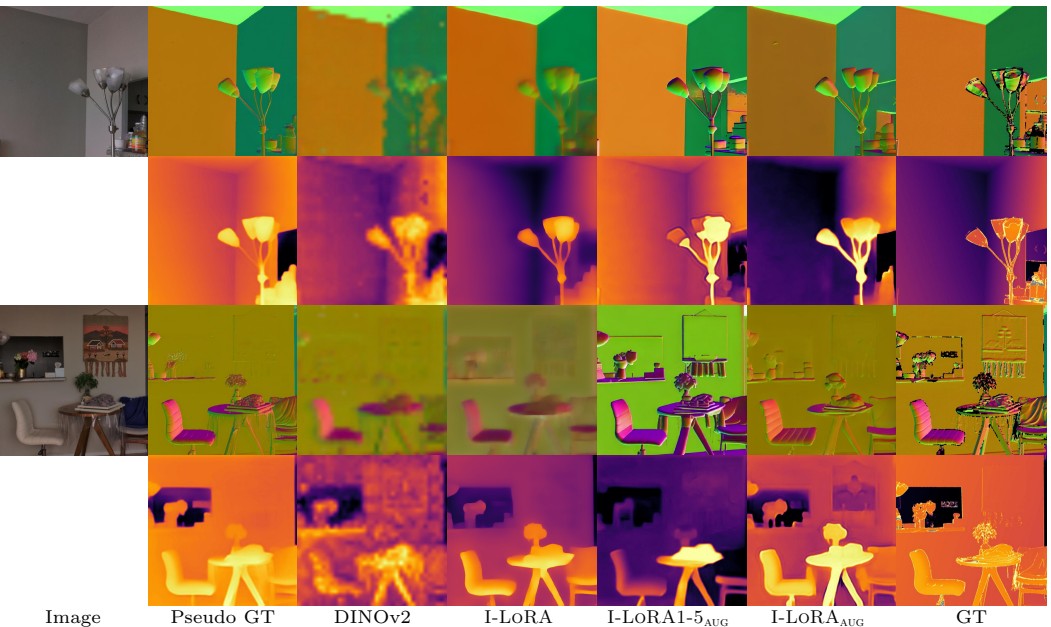

| Image | Pseudo GT | DINOv2 | I-LoRA | I-LoRA1-5$_{\text{AUG}}$ | I-LoRA$_{\text{AUG}}$ | GT |

Figure 12: We show normals (top in each set) and depth (bottom in each set) derived from improved multi-step diffusion process from our I-LoRA$_{\text{AUG}}$ compared with other alternatives. I-LoRA1-5$_{\text{AUG}}$ is similar to I-LoRA$_{\text{AUG}}$ except it uses SDv1-5 and does not use Zero SNR strategy. I-LoRA1-5$_{\text{AUG}}$ presents sharper details, especially in complex areas – structures like lamp stand and chair. I-LoRA$_{\text{AUG}}$, on the other hand, illustrates a significant improvement in reducing color shifts while maintaining sharpness, as seen in the comparison with ground truth in the last column.

Our approach shows that models can generate intrinsic images directly from the same decoder head, despite these images being out-of-distribution and unlike real images. Surprisingly, it is easy to extract this information using our method, which is broadly applicable across generative models, supporting our argument that these representations are indeed inherent in the learned model.

**Limitations**. Although we have demonstrated that generative models carry a wealth of intrinsic information, it is still ambiguous how these models use this information when generating images. Secondly, even though I-LoRA is both parameter and label-efficient, we believe there is still room for further reduction of training requirements and perhaps the development of a parameter-free approach. Lastly, the I-LoRA$_{\text{AUG}}$ generates sharper results but still lags behind its single-step counterpart in terms of quantitative analysis. Further work is needed to explore this question.

Our **future work** will focus on overcoming the limitations we have encountered so far and expanding on our findings. One way to do this is by explicitly incorporating the extracted scene intrinsics into the learning process of generative image models to improve them further via efficient fine-tuning. Additionally, developing an evaluation of generative models based on physical properties may help develop interpretable metrics. Another direction interesting to explore is learning specific latent codes similar to Bhattad et al. (2023a) for each intrinsics, as opposed to using LoRA modules, to further reduce the number of learned parameters.

**Broader Impact Statement.** Our paper introduces "INTRINSIC LoRA", a model-agnostic framework for extracting intrinsic properties and visual knowledge from generative models. This advancement has the potential to significantly enhance the interpretability and usability of generative models across various applications, from computer vision to autonomous systems. By enabling efficient and accurate extraction of scene intrinsics, our approach could improve image generation quality and provide deeper insights into how these models understand and recreate the visual world. Furthermore, the ability to recover intrinsic information with minimal data and computational resources promotes more sustainable and accessible AI research and development. Finally, our work contributes to the foundational understanding of generative models, paving the way for more responsible, transparent and effective AI systems in the future.

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

# A  Additional Ablation Studies

## A.1  Number of Diffusion Steps

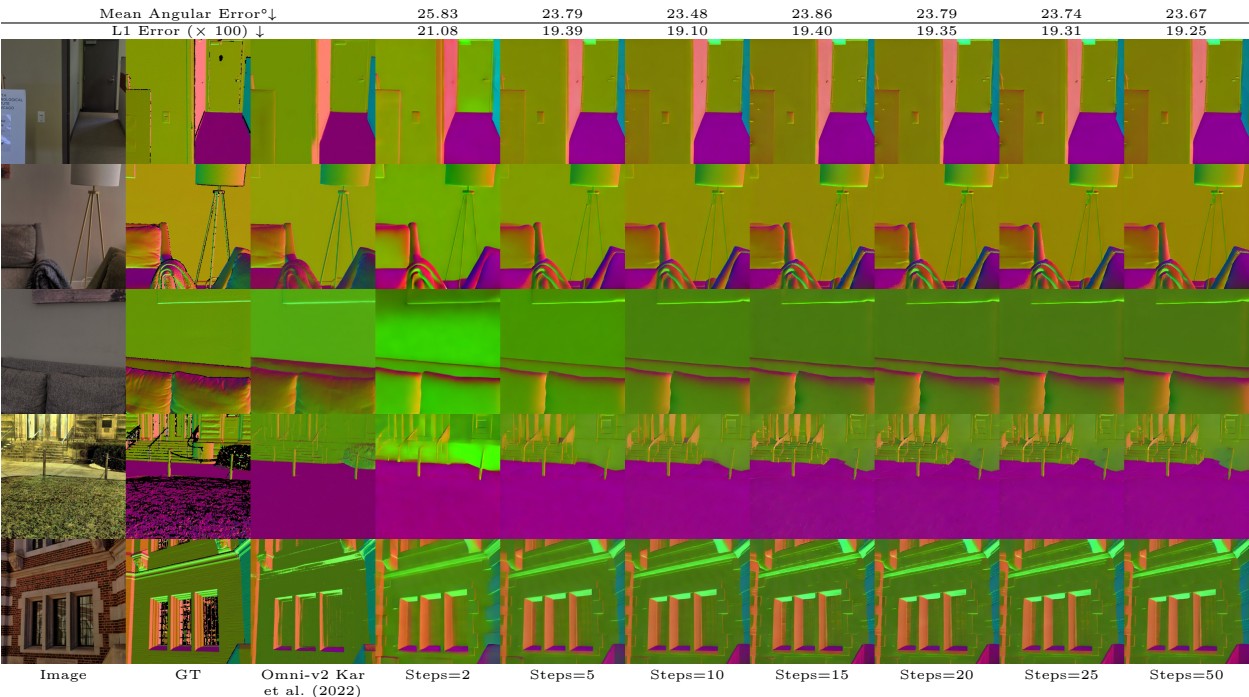

| Mean Angular Error°↓ | | | 25.83 | 23.79 | 23.48 | 23.86 | 23.79 | 23.74 | 23.67 |
| L1 Error (× 100) ↓ | | | 21.08 | 19.39 | 19.10 | 19.40 | 19.35 | 19.31 | 19.25 |
| Image | GT | Omni-v2 Kar et al. (2022) | Steps=2 | Steps=5 | Steps=10 | Steps=15 | Steps=20 | Steps=25 | Steps=50 |

Figure 13: Ablation study to determine the effect of varying numbers of diffusion steps while keeping CFG fixed at 3.0. Our findings show that there are very small differences, both in terms of quantity and quality, after 10 steps. For our main paper, we report results for 25 steps as it is more stable across different intrinsics.

To assess the impact of the number of diffusion steps on the performance of the multi-step I-LoRA$_\text{AUG}$ model, we conducted an ablation study. The results are presented in Fig. 13. For all our experiments in the main text, we used DPMSolver++ (Lu et al., 2022). Interestingly, the quality of results did not vary significantly with an increased number of steps, indicating that 10 steps are sufficient for extracting better surface normals from the Stable Diffusion. Nevertheless, we use 25 steps for all our experiments because it is more stable across different image intrinsics.

## A.2  CFG scales

When working with the multi-step I-LoRA$_\text{AUG}$, the quality of the final output is influenced by the choice of classifier-free guidance (CFG) scales during the inference process. In Fig. 14, we present a comparison of the effects of using different CFG scales. Based on our experiments, we found that using CFG=3.0 results in the best overall quality and minimizes color-shift artifacts.

# B  Other Ablations and Baselines

We extensively study the effect of applying LoRA to different attention layers within Stable Diffusion models. Specifically, we investigate the outcomes of targeting up-blocks, mid-block, down-blocks, cross-attention, and self-attention layers individually. We find (Fig. 15) that isolating LoRA to up or down blocks or the mid-block alone is less effective or diverges, and applying to either cross- or self-attention layers yields decent results, though combining them is best.

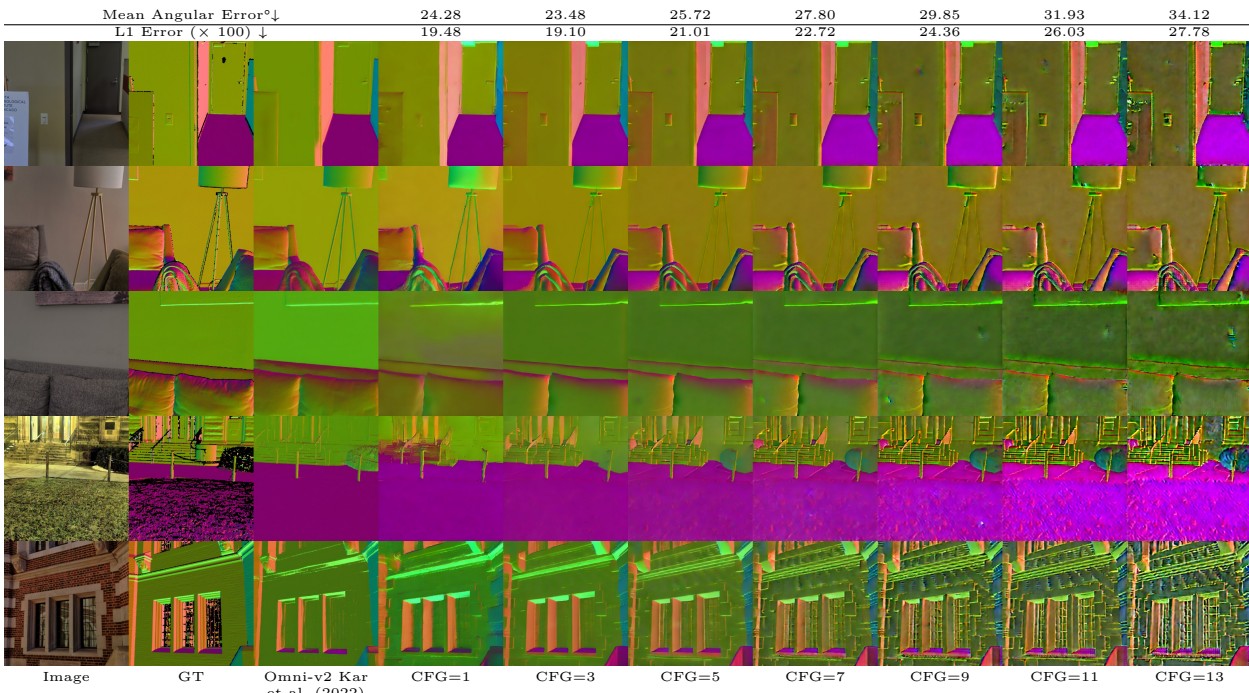

| Mean Angular Error°↓ | | | 24.28 | 23.48 | 25.72 | 27.80 | 29.85 | 31.93 | 34.12 |
| L1 Error (× 100) ↓ | | | 19.48 | 19.10 | 21.01 | 22.72 | 24.36 | 26.03 | 27.78 |
| Image | GT | Omni-v2 Kar et al. (2022) | CFG=1 | CFG=3 | CFG=5 | CFG=7 | CFG=9 | CFG=11 | CFG=13 |

Figure 14: Ablation study analyzing the impact of different classifier-free guidance (CFG) on I-LoRA$_{\text{AUG}}$ surface normal prediction. For efficiency, we experimented with a step of 10. We observed that CFG=1 sometimes led to incorrect semantic predictions, particularly in the case of stairs in row 4. On the other hand, using large CFGs (5 and beyond) results in more severe color shift problems.

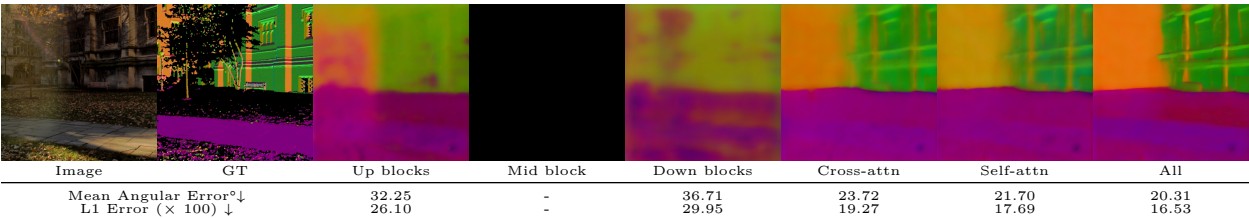

| | Image | GT | Up blocks | Mid block | Down blocks | Cross-attn | Self-attn | All |
| --- | --- | --- | --- | --- | --- | --- | --- | --- |
| Mean Angular Error°↓ | | | 32.25 | - | 36.71 | 23.72 | 21.70 | 20.31 |
| L1 Error (× 100) ↓ | | | 26.10 | - | 29.95 | 19.27 | 17.69 | 16.53 |

Figure 15: Ablation study on the effect of applying LoRA on different types of attention layers. We started all models with SD v1-5, 4000 training samples and LoRA rank=8. Training with LoRA only on the mid block never converges.

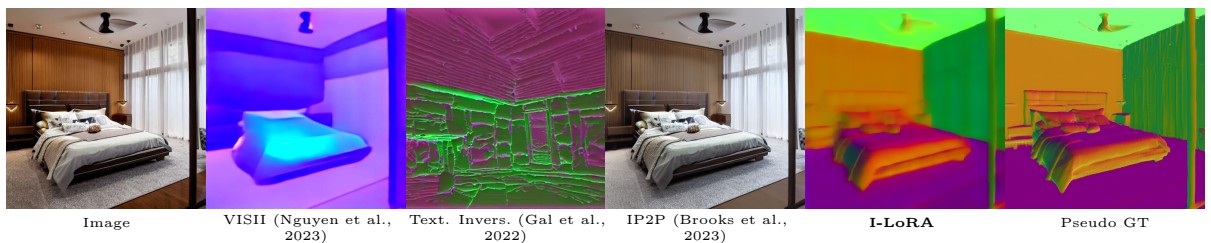

| Image | VISII (Nguyen et al., 2023) | Text. Invers. (Gal et al., 2022) | IP2P (Brooks et al., 2023) | **I-LoRA** | Pseudo GT |

Figure 16: Comparison of image editing techniques for surface normal mapping. VISII and Textual Inversion yield unsatisfactory results, while InstructPix2Pix fails to interpret the task, resulting in near-original output.

Additionally, we evaluated other image editing methods such as Textual Inversion (Gal et al., 2022) and VISII (Nguyen et al., 2023), alongside InstructPix2Pix's response to "Turn it into a surface normal map"

Table 5: Comparison of quality of normals extracted from StyleGAN Bhattad et al. (2023a).

|  | Mean Error°↓ | Median Error°↓ | L1 × 100 ↓ |
|---|---|---|---|
| "StyleGAN knows" (Bhattad et al., 2023a) | 19.92 | 46.65 | 16.64 |
| I-LoRA-StyleGAN (Ours) | **13.24** | **23.55** | **10.92** |

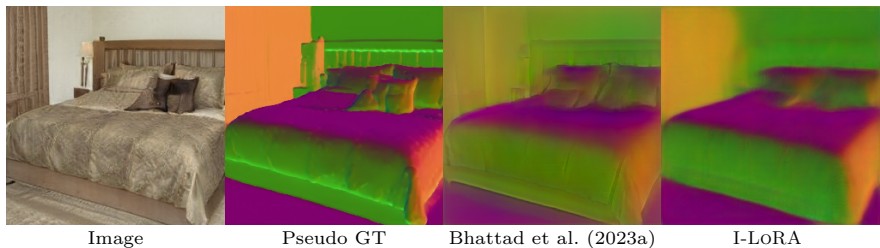

| Image | Pseudo GT | Bhattad et al. (2023a) | I-LoRA |

Figure 17: Qualitative results of normals extracted from StyleGAN by Bhattad et al. (2023a) and Ours.

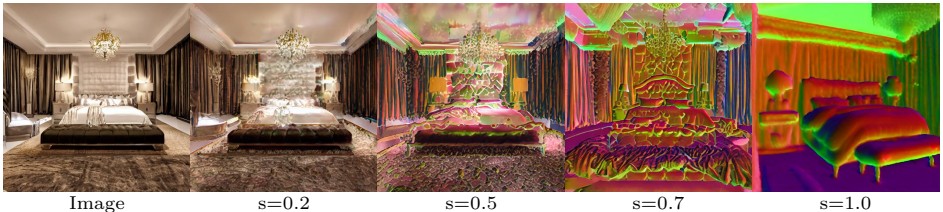

| Image | s=0.2 | s=0.5 | s=0.7 | s=1.0 |

Figure 18: We observe applying SDEdit method on the SDv1-5 model alone, without incorporating the additional input image latent encoding, fails to produce satisfactorily aligned and high-quality scene intrinsics. The reason for this might be the considerable domain shift that exists between RGB images and surface normal maps, which results in severe artifacts when using SDEdit. The variable "s" represents the strength of SDEdit.

instruction (Brooks et al., 2023). As shown in Fig. 16, these methods perform poorly for intrinsic image extraction, demonstrating the effectiveness of our I-LoRA approach in extracting scene intrinsics.

We also provide a comparison with Bhattad et al. (2023a) in Tab. 5 and Fig. 17. This comparison is for the same 500 randomly generated images. I-LoRA outperforms Bhattad et al. (2023a) significantly.

In addition, we show that directly applying SDEdit (Meng et al., 2021) will also fail to extract reasonable image intrinsics. We take the model from the SDv1-5 column in Fig.13 of the main paper and apply SDEdit. In Fig. 18, we show directly applying SDEdit results in severe artifacts, regardless of strength.

## C  Hyper-parameters

In Table 6, we show the hyperparameters we use for each model.

## D  Generated Images Used for Quantitative Analysis

In Tab. 2 of the main paper, we report quantitative results on synthetic images. For Autoregressive models and GANs, we first randomly sample 500 noises and use them to generate 500 RGB images. The same 500 noises will then be used to generate intrinsics with our learned LoRAs loaded. For Stable Diffusion experiments (both single-step and multi-step), we use a single dataset with 1000 synthetic images with various prompts.

The pseudo GT are obtained by applying SOTA off-the-shelf models on the RGB images.

Table 6: Hyper-parameters for each model. LR refers to the learning rate and BS refers to the batch size. Please note that the number of steps required to reach convergence reported above is for normal/depth. However, it is worth noting that albedo and shading tend to require significantly fewer steps to converge (usually half of normal/depth). Additionally, I-LoRA$_{\text{AUG}}$ (multi-step) and I-LoRA (single-step) are trained on real-world DIODE dataset, while the other models are trained on synthetic images within a specific domain. (Num. of params of VQGAN counts transformer + first stage models; Num. of params of I-LoRA$_{\text{AUG}}$ and I-LoRA counts VAE+UNet)

| Model | Dataset | Resolution | Rank | LR | BS | LoRA Params | Generator Params | Convergence Steps |
|---|---|---|---|---|---|---|---|---|
| VQGAN | FFHQ | 256 | 8 | 1e-03 | 1 | 0.13M | 873.9M | $\sim 4000$ |
| StyleGAN-v2 | FFHQ | 256 | 8 | 1e-03 | 1 | 0.14M | 24.8M | $\sim 4000$ |
| StyleGAN-v2 | LSUN Bedroom | 256 | 8 | 1e-03 | 1 | 0.14M | 24.8M | $\sim 4000$ |
| StyleGAN-XL | FFHQ | 256 | 8 | 1e-03 | 1 | 0.19M | 67.9M | $\sim 4000$ |
| StyleGAN-XL | ImageNet | 256 | 8 | 1e-03 | 1 | 0.19M | 67.9M | $\sim 4000$ |
| I-LoRA$_{\text{AUG}}$ (multi step) | Open | 512 | 8 | 1e-04 | 4 | 1.59M | 943.2M | $\sim 30000$ |
| I-LoRA (single step) | Open | 512 | 8 | 1e-04 | 4 | 1.59M | 943.2M | $\sim 15000$ |

## E  Additional Qualitative Results

In Fig. 19, we present more results for I-LoRA$_{\text{AUG}}$ and I-LoRA1-5$_{\text{AUG}}$ . Fig. 20 shows extra results for models trained on FFHQ dataset. More examples of scene intrinsics extracted from StyleGAN-v2 trained on LSUN bedroom can be found in Fig. 21. In Fig. 22, we show results for Stable Diffusion I-LoRA (single-step) on generated images. Shown in Fig. 23 are extra results for StyleGAN-XL trained on ImageNet.

## F  Results on $1024^2$ synthetic images

Our multi-step I-LoRA$_{\text{AUG}}$ models, although trained exclusively on $512^2$ images from the DIODE dataset, demonstrate their robustness by successfully extracting intrinsic images from $1024^2$ high-resolution synthetic images generated by Stable Diffusion XL (Podell et al., 2023), as shown across Figures 24 to 33

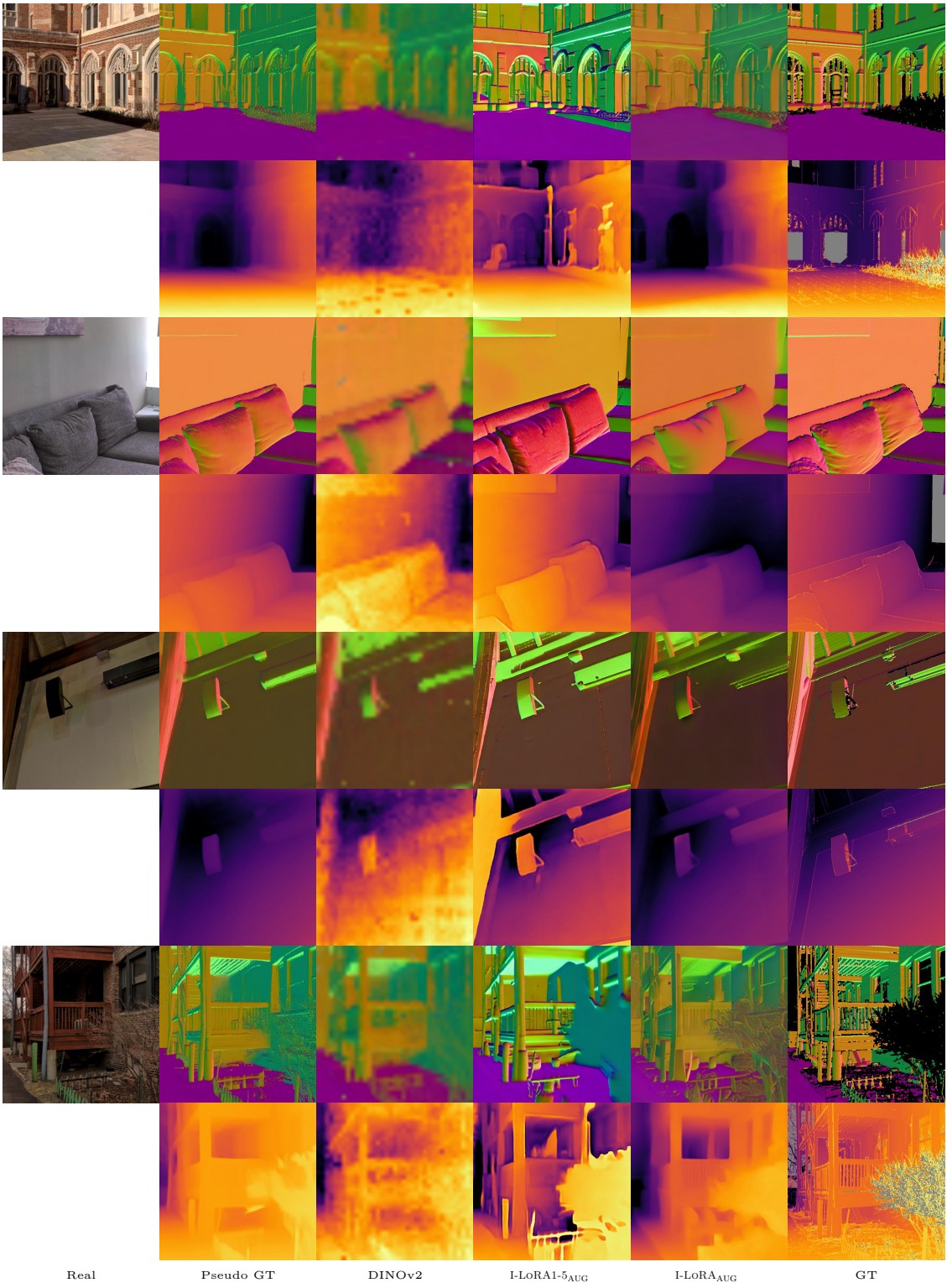

Real   Pseudo GT   DINOv2   I-LoRA1-5$_{\text{AUG}}$   I-LoRA$_{\text{AUG}}$   GT

Figure 19: Additional results after applying improved diffusion techniques with I-LoRA$_{\text{AUG}}$. I-LoRA$_{\text{AUG}}$ was found to significantly reduce color shift artifacts observed in I-LoRA1-5$_{\text{AUG}}$ during the extraction of detailed scene intrinsic results.

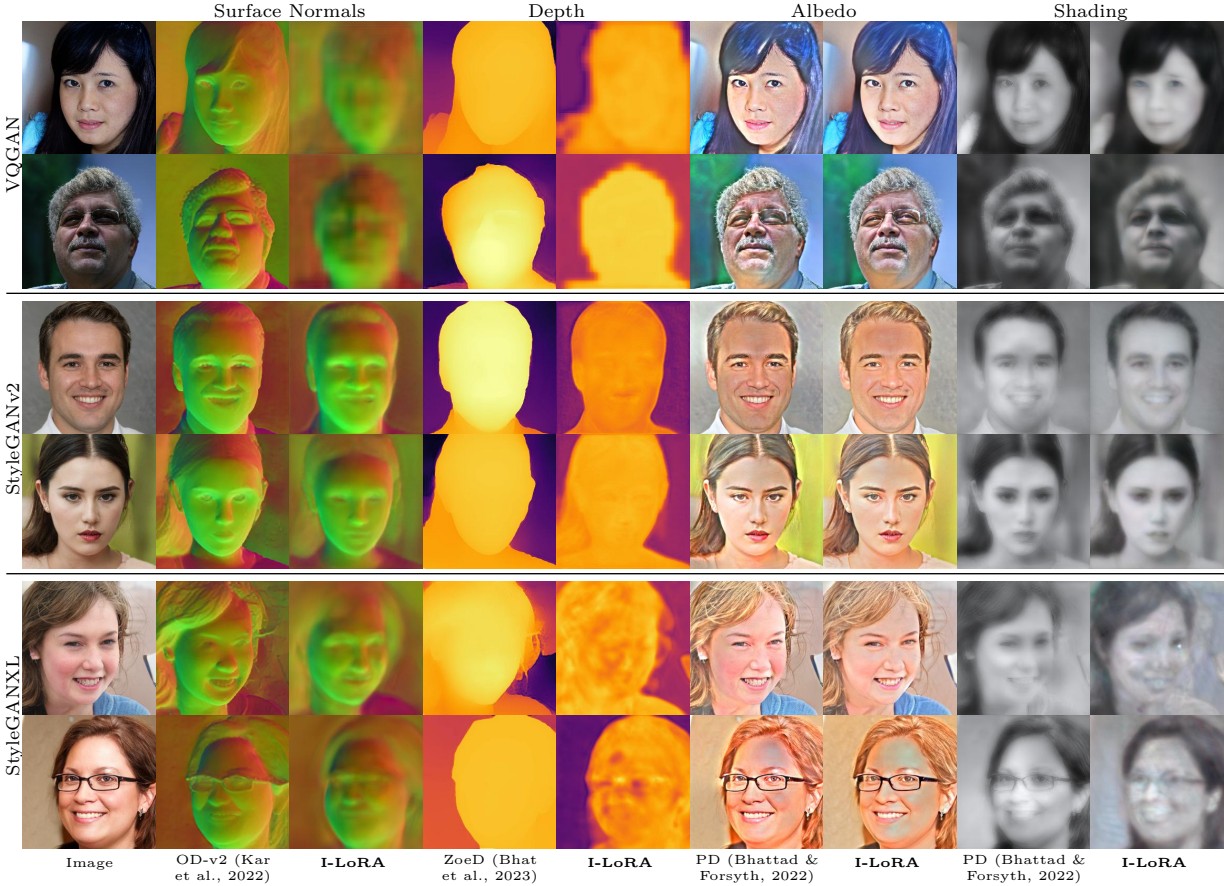

Figure 20: Additional results of scene intrinsics from different generators – VQGAN, StyleGAN-v2, and StyleGAN-XL – trained on FFHQ dataset.

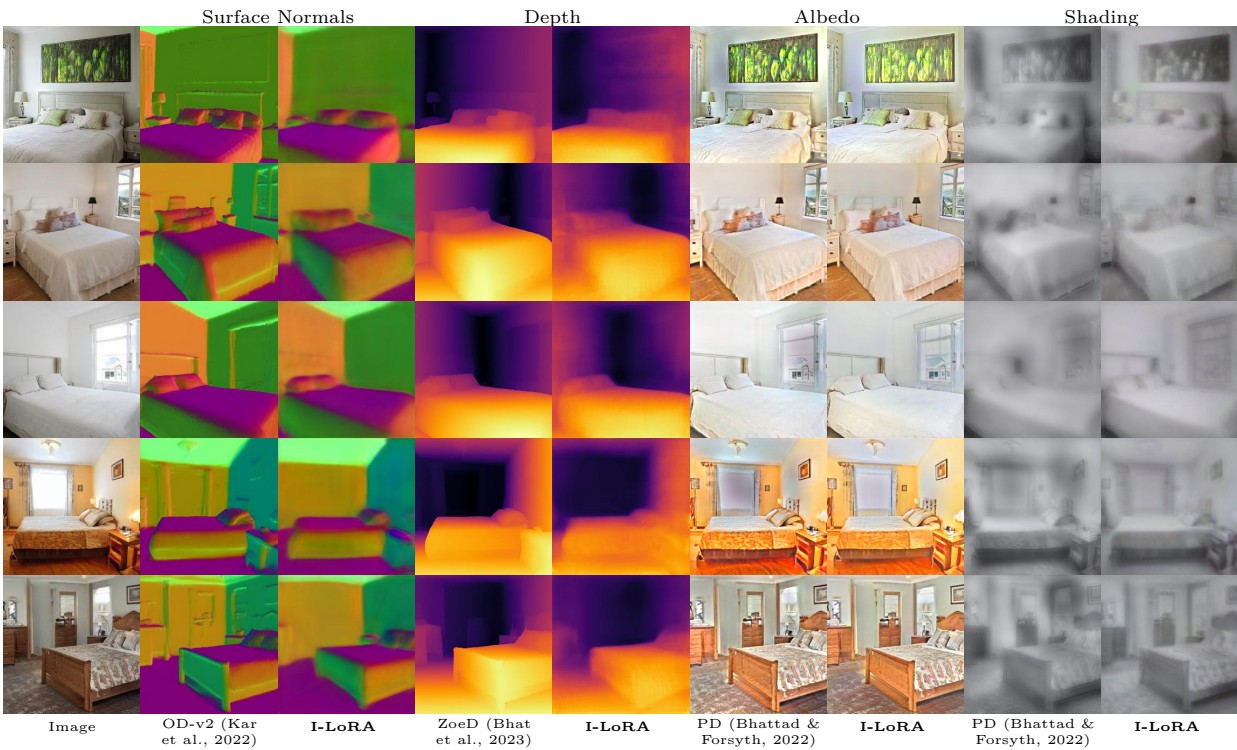

Figure 21: Additional results of scene intrinsics extraction from Stylegan-v2 trained on LSUN bedroom images.

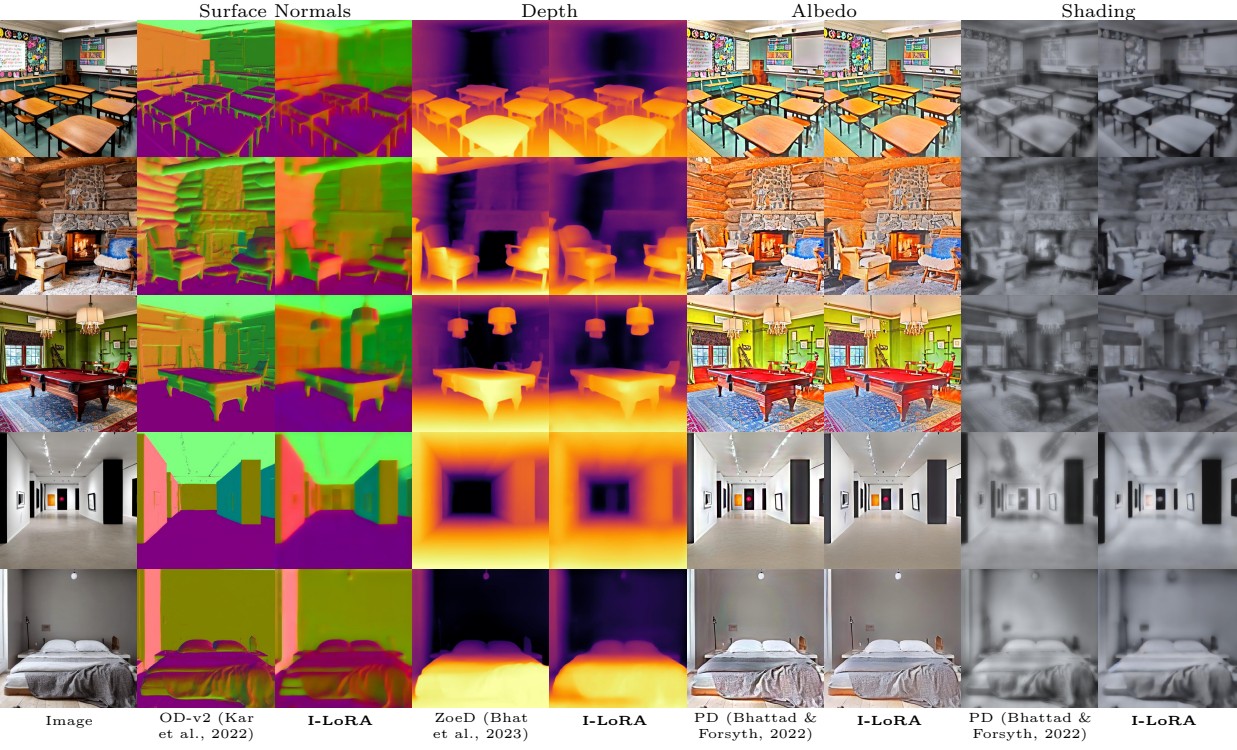

Figure 22: Additional results of scene intrinsics extraction from Stable Diffusion I-LoRA (single-step).

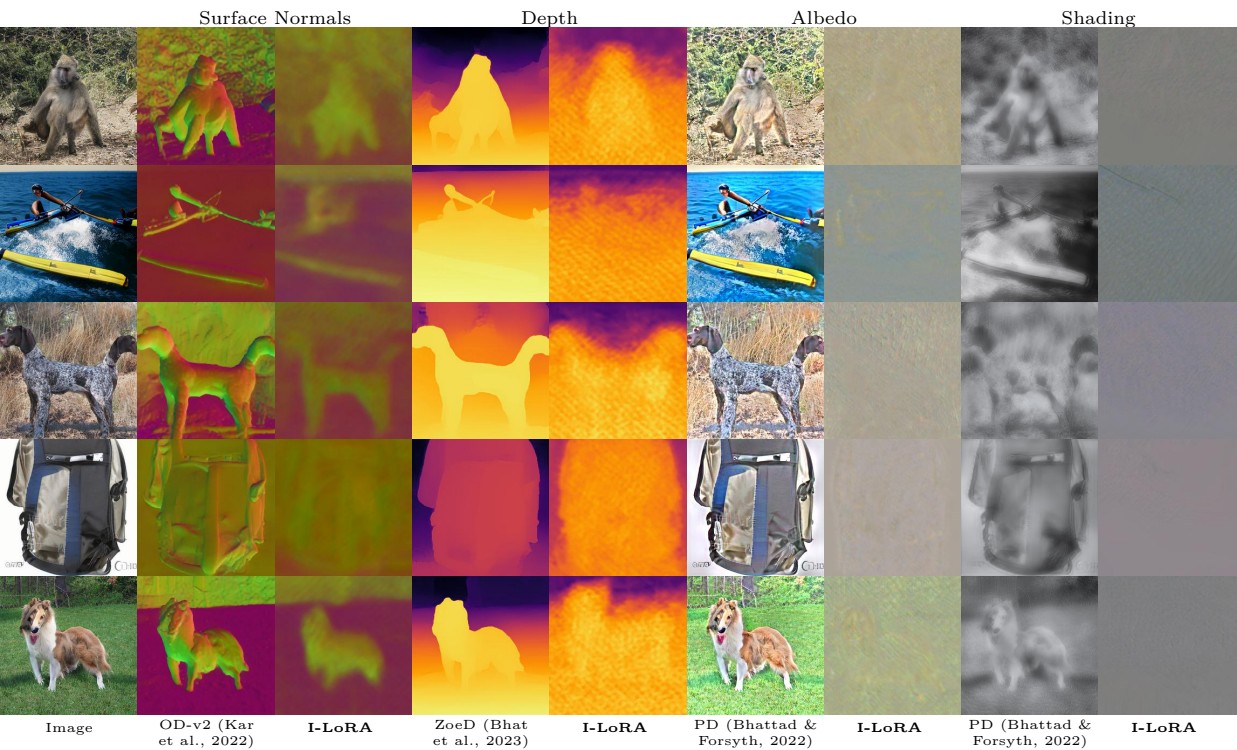

Figure 23: Additional results for StyleGAN-XL trained on ImageNet. StyleGAN-XL's inability to produce image intrinsics may be due to its inability to create high-quality plausible images.

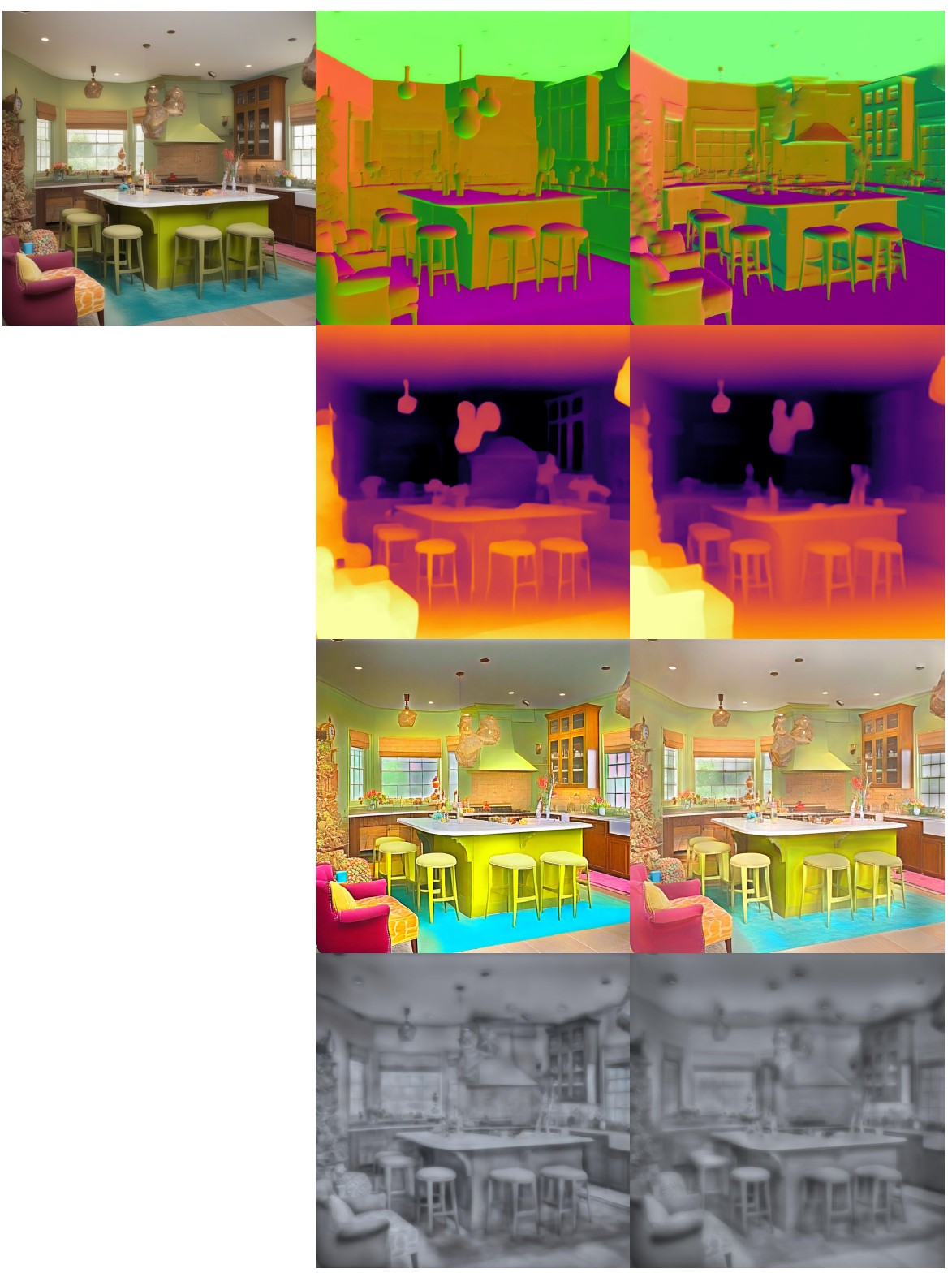

Figure 24: Results of I-LoRA$_{\text{AUG}}$ models applied on unseen $1024^2$ synthetic images. Left: original image; middle: ours; right: pseudo ground truth.

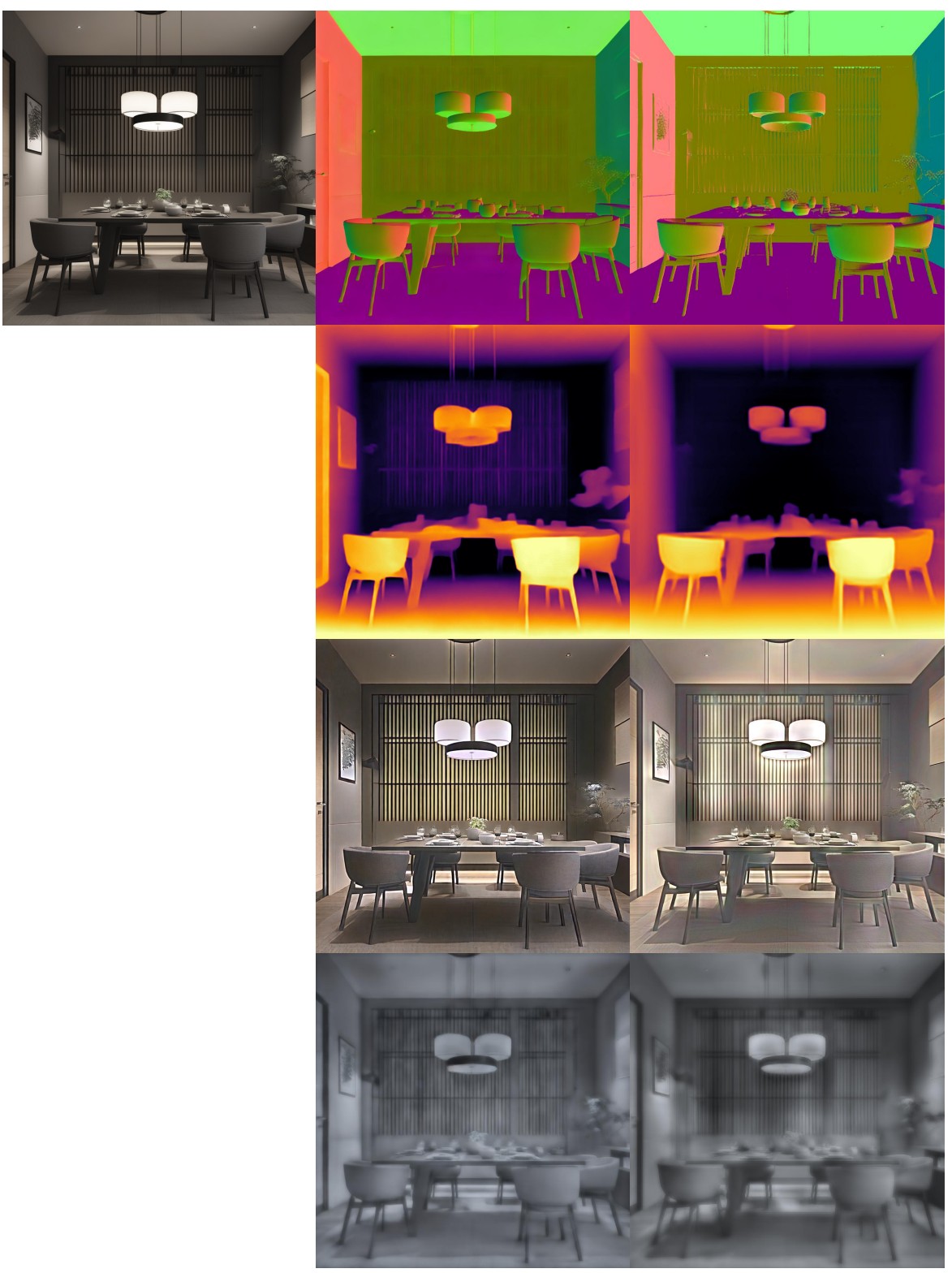

Figure 25: Cont. results of I-LoRA$_{\text{AUG}}$ models applied on unseen $1024^2$ synthetic images. Left: original image; middle: ours; right: pseudo ground truth.

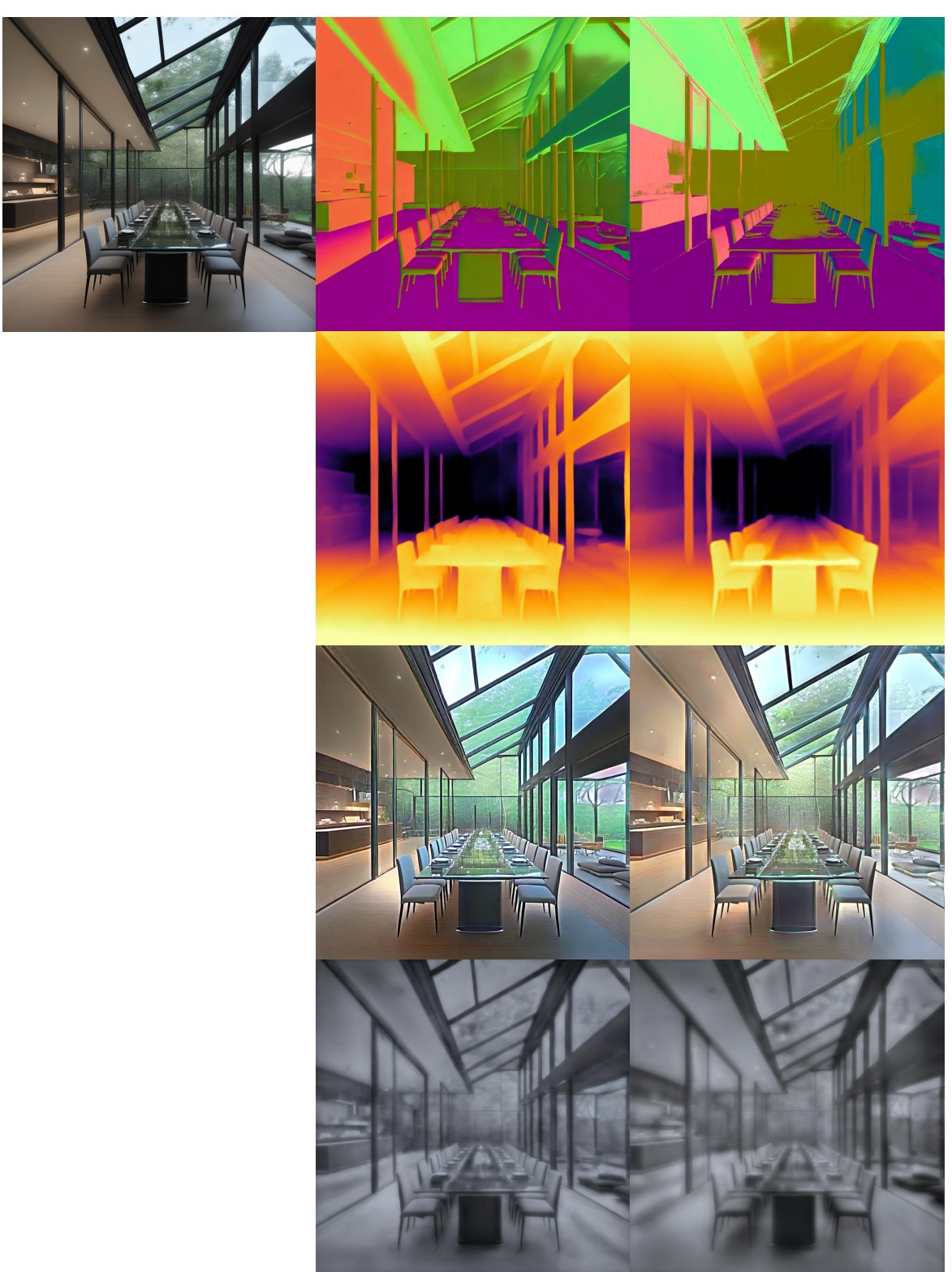

Figure 26: Cont. results of I-LoRA$_{\text{AUG}}$ models applied on unseen $1024^2$ synthetic images. Left: original image; middle: ours; right: pseudo ground truth.

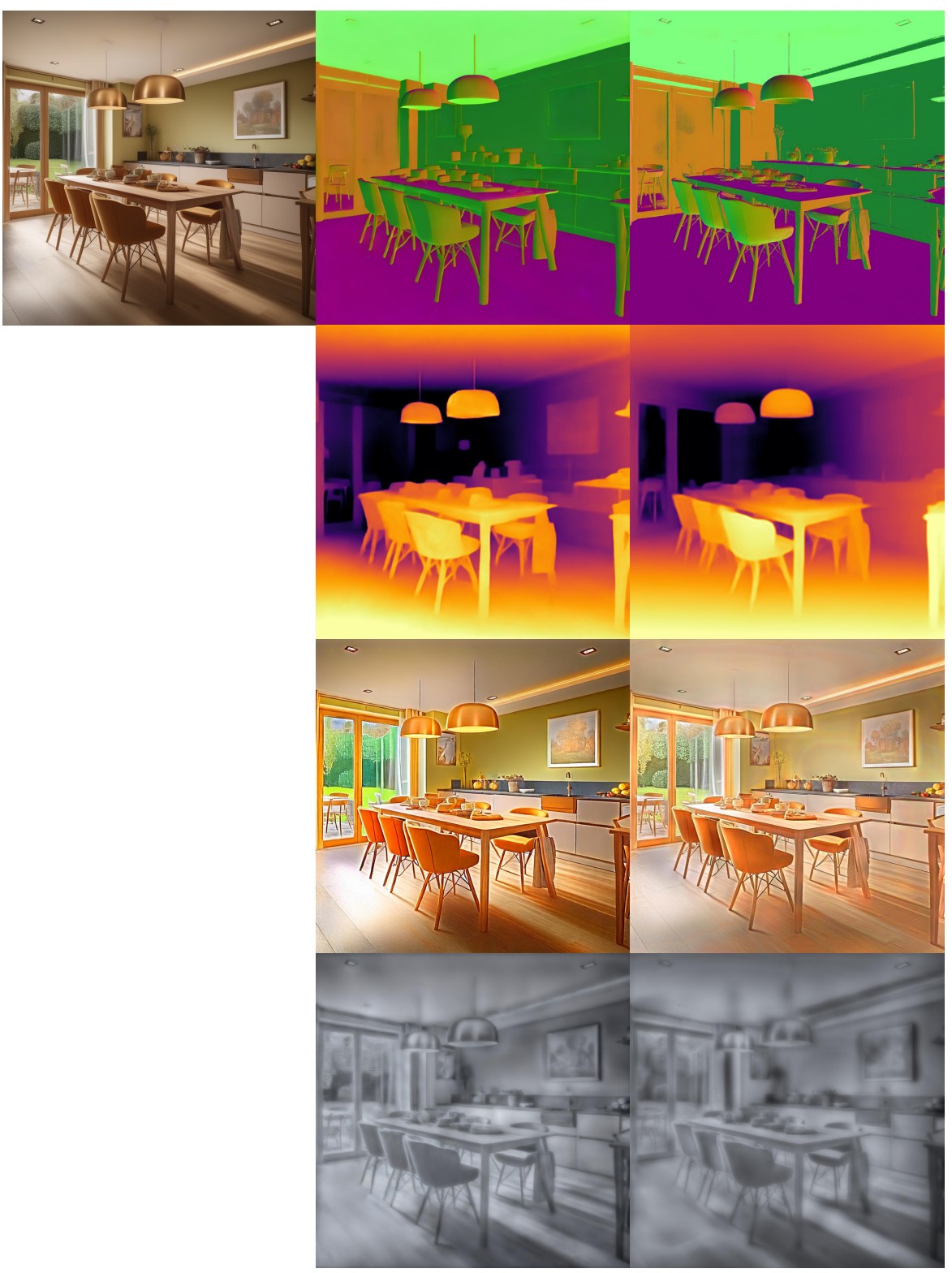

Figure 27: Cont. results of I-LoRA$_{\text{AUG}}$ models applied on unseen $1024^2$ synthetic images. Left: original image; middle: ours; right: pseudo ground truth.

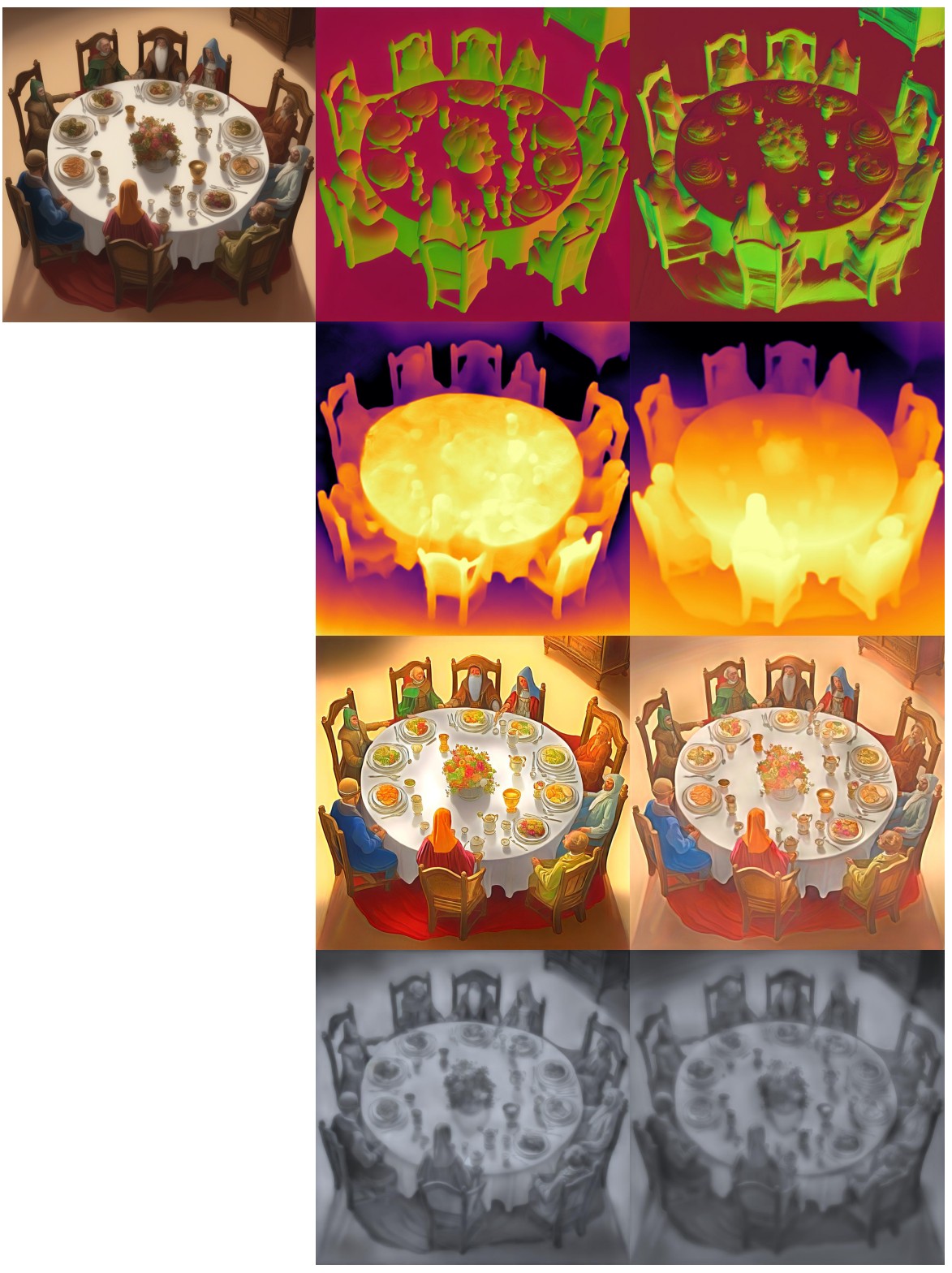

Figure 28: Cont. results of I-LoRA$_{\text{AUG}}$ models applied on unseen $1024^2$ synthetic images. Left: original image; middle: ours; right: pseudo ground truth.

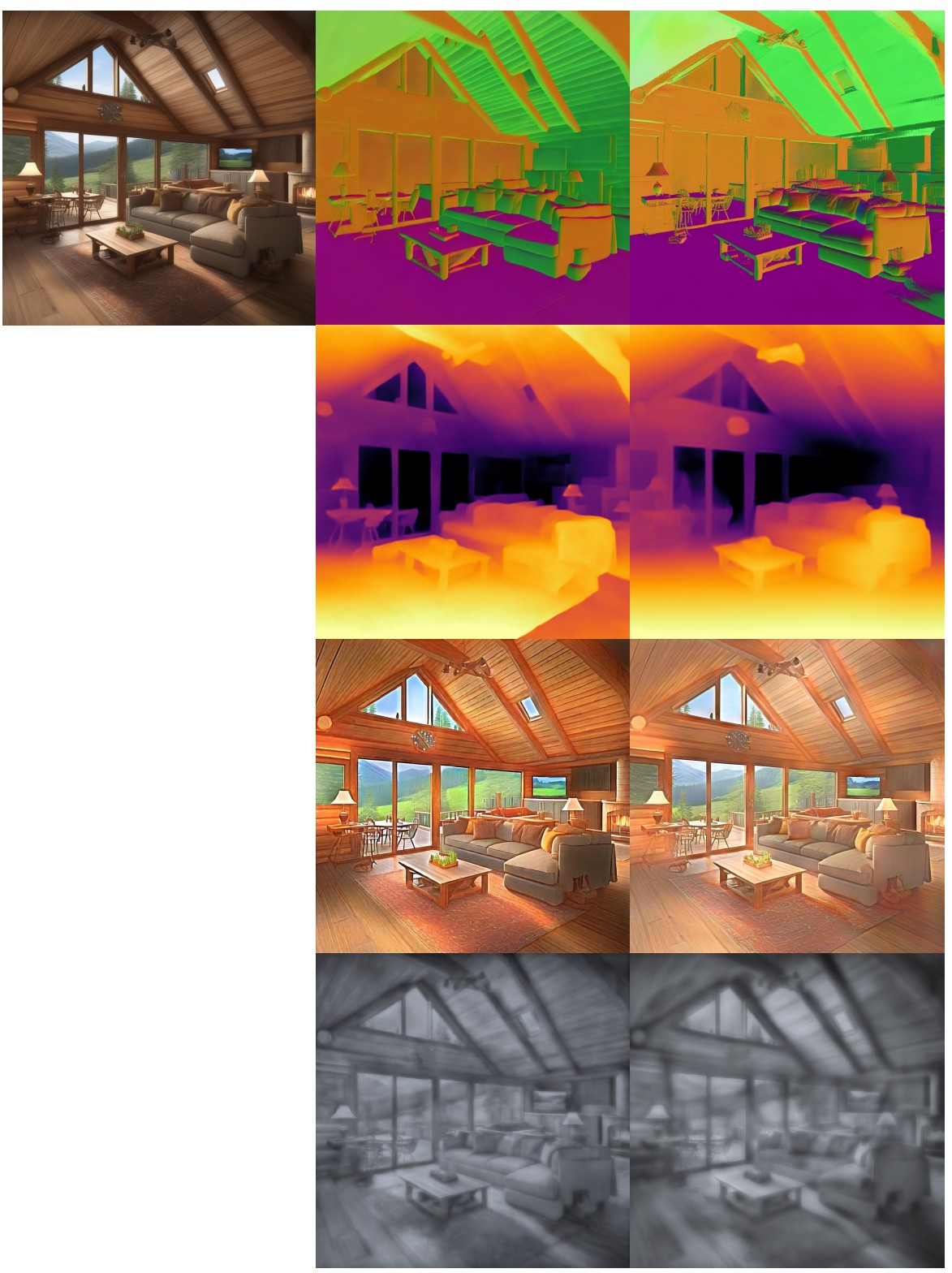

Figure 29: Cont. results of I-LoRA$_{\textsc{aug}}$ models applied on unseen $1024^2$ synthetic images. Left: original image; middle: ours; right: pseudo ground truth.

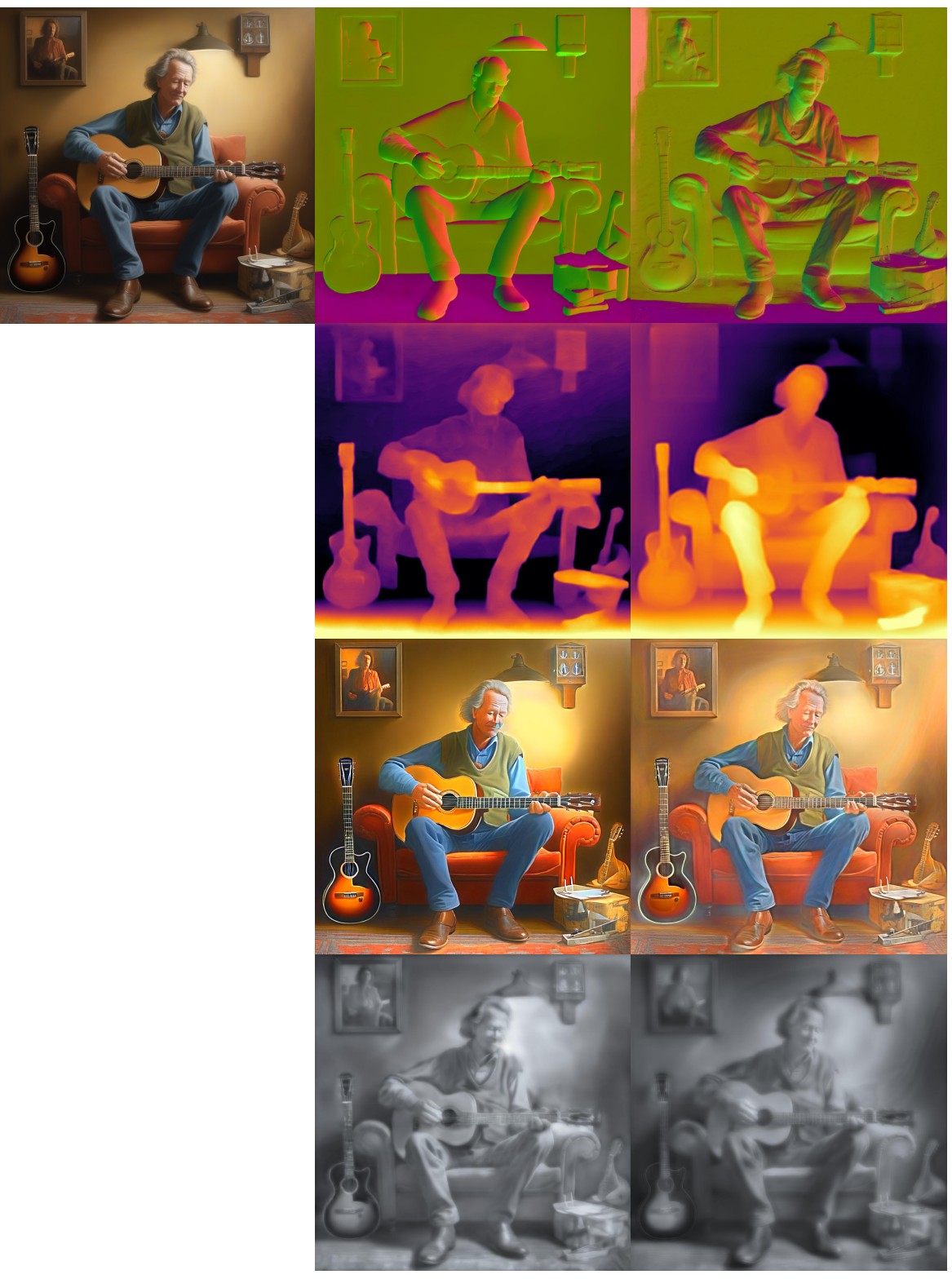

Figure 30: Cont. results of I-LoRA$_{\text{AUG}}$ models applied on unseen $1024^2$ synthetic images. Left: original image; middle: ours; right: pseudo ground truth.

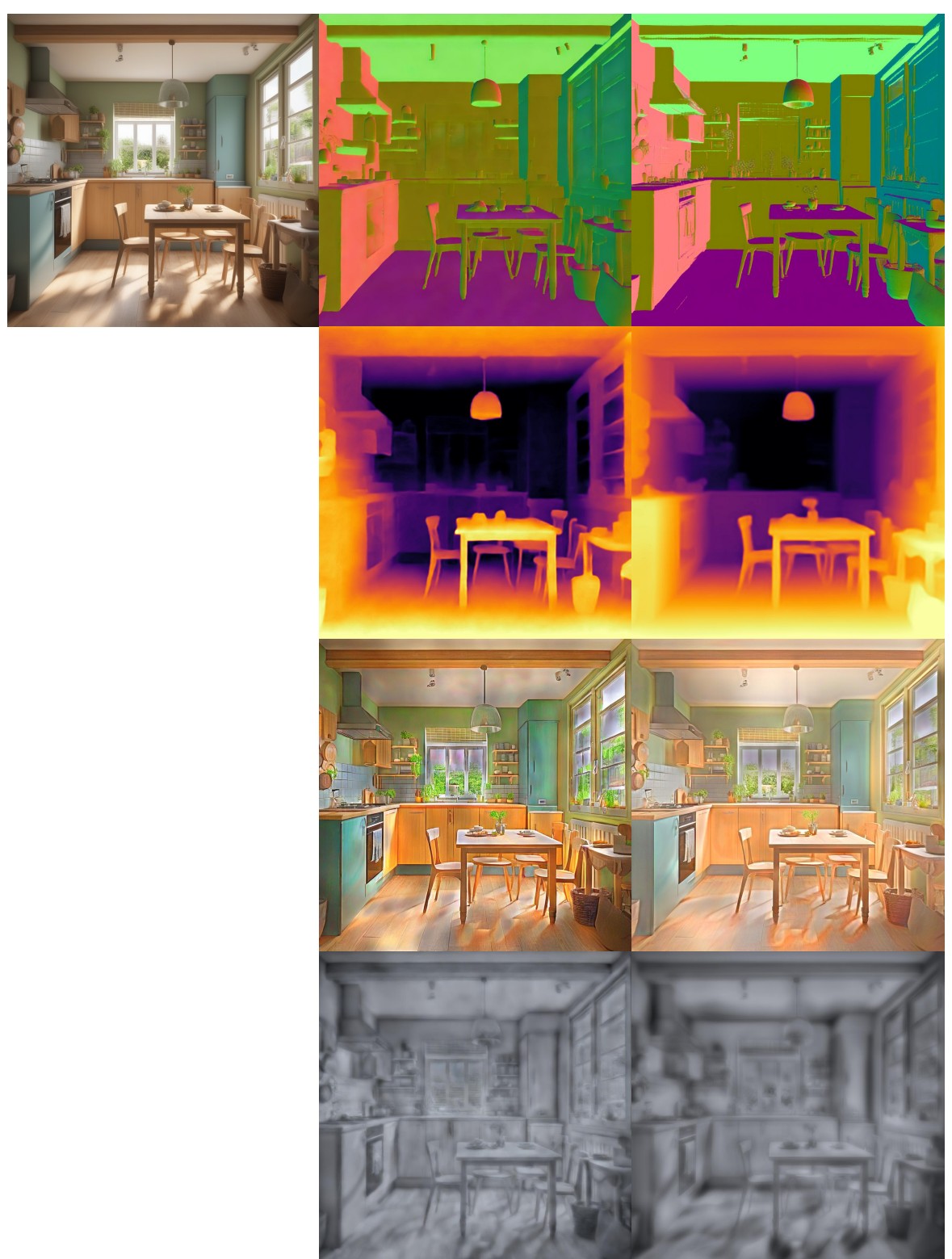

Figure 31: Cont. results of I-LoRA$_{\text{AUG}}$ models applied on unseen $1024^2$ synthetic images. Left: original image; middle: ours; right: pseudo ground truth.

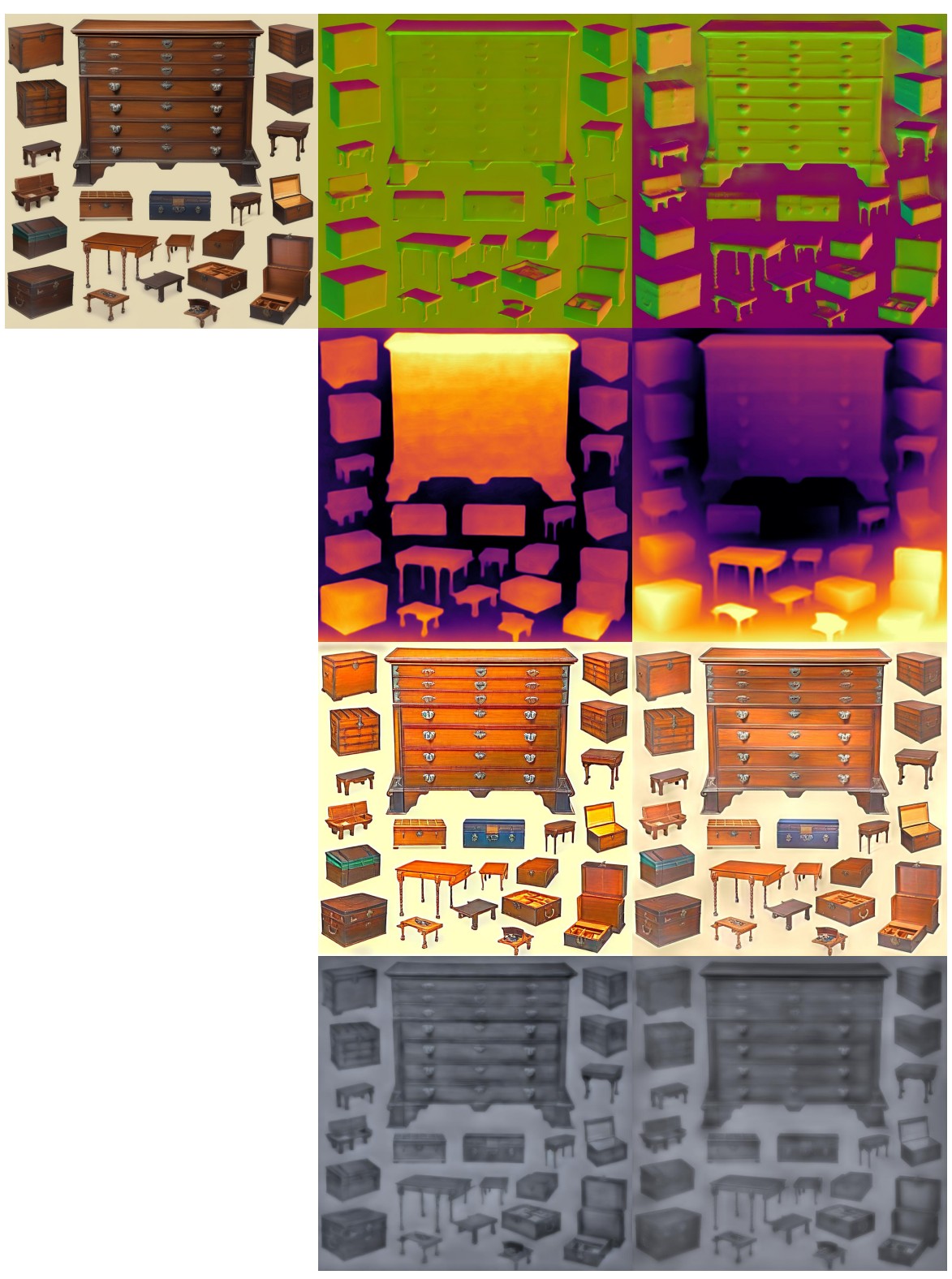

Figure 32: Cont. results of I-LoRA$_{\text{AUG}}$ models applied on unseen $1024^2$ synthetic images. Left: original image; middle: ours; right: pseudo ground truth.

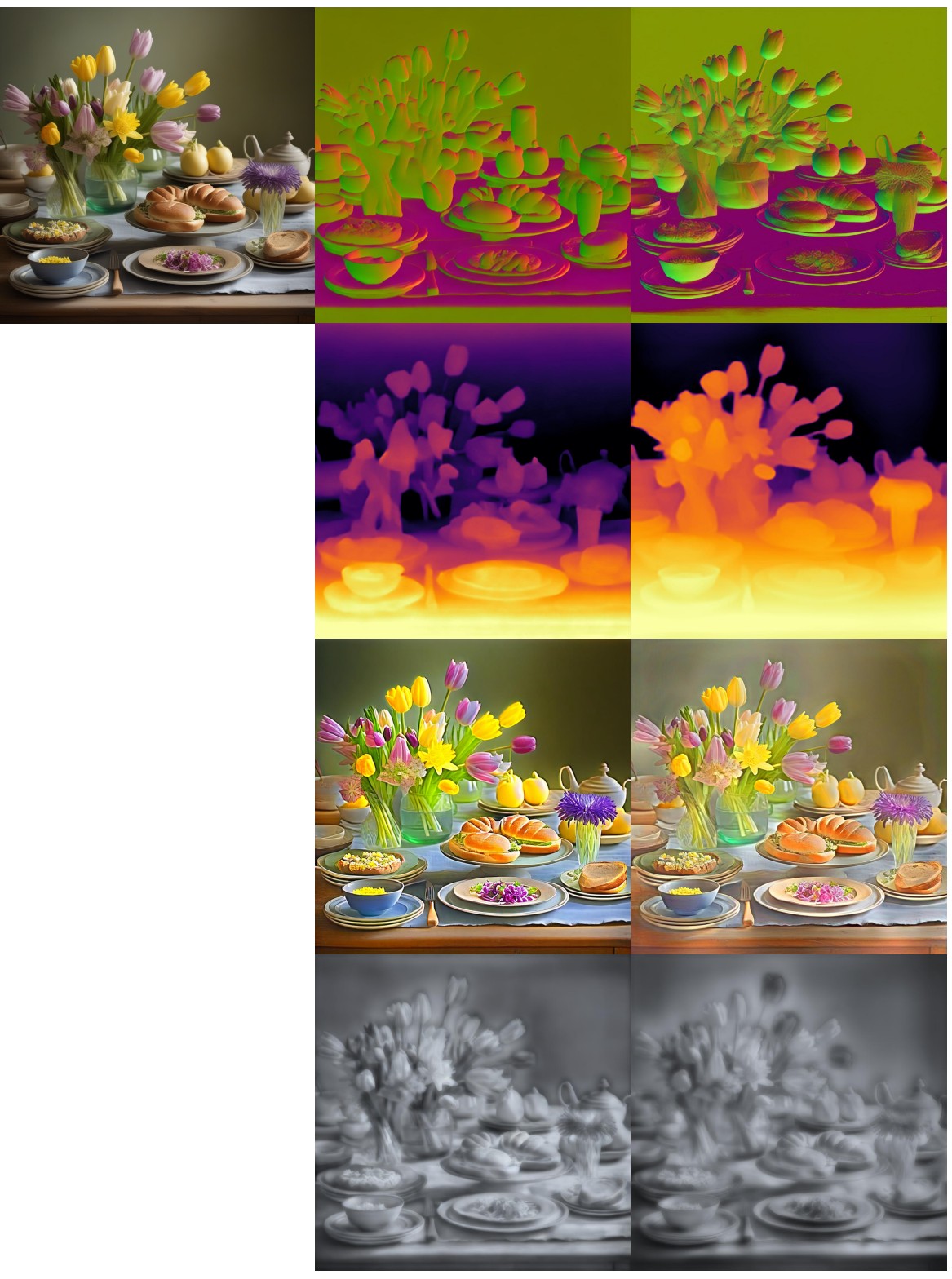

Figure 33: Cont. results of I-LoRA$_{\text{AUG}}$ models applied on unseen $1024^2$ synthetic images. Left: original image; middle: ours; right: pseudo ground truth.

