# OpenReview forum: "Generative Models: What Do They Know? Do They Know Things? Let's Find Out!"
_TMLR — Rejected by TMLR_

### Review · Reviewer_YCyC · 2024-07-27

**Summary Of Contributions:**

The paper introduces Intrinsic LoRA (I-LoRA), a method for extracting scene intrinsics such as normals, depth, albedo, and shading from various generative models, including Diffusion models, GANs, and Autoregressive models. I-LoRA leverages Low-Rank Adaptation (LoRA) to fine-tune specific layers of generative models with minimal computational overhead and small datasets. The approach is model-agnostic, requiring less than 0.17% additional parameters and as few as 250 labeled images. The paper demonstrates that I-LoRA can extract high-quality intrinsics comparable to those obtained from supervised methods trained on millions of examples. The findings indicate that the intrinsic knowledge within generative models is a byproduct of the large-scale learning process, with the quality of extracted intrinsics correlating with the visual quality of the generative model's outputs.

**Audience:**

Yes

**Claims And Evidence:**

No

**Requested Changes:**

I feel the paper needs a more clear positioning since its technical components highly overlap LoRA. It is a bit unfair to call it a new method I-LoRA just because it is applied to a new prediction task.

**Strengths And Weaknesses:**

Strengths:
- The paper performed a systematic study on applying LoRA to image intrinsics generation, showing that various image generative models have an inherent understanding of image intrinsics which can be extracted by training on a small amount of data with LoRA.
- The paper is well-written and easy to follow.
- Results demonstrated that the proposed approach outperforms baselines like linear probing.
- Code is provided for reproducibility.

Weaknesses:
- The paper lacks technical novelty and mainly applies LoRA to image generation models for intrinsics.
- The finetuning baseline needs to be more carefully designed. It is no surprise that finetuning on a small number of image intrinsics won't work well since the model quickly overfits and forgets about the prior knowledge. Combining the dataset with image data and sharing different modalities as in [1] or other more feasible approaches [2] will make a stronger baseline.


[1] Huang, Xin, et al. "Humannorm: Learning normal diffusion model for high-quality and realistic 3d human generation." Proceedings of the IEEE/CVF Conference on Computer Vision and Pattern Recognition. 2024.
[2] Ke, Bingxin, et al. "Repurposing diffusion-based image generators for monocular depth estimation." Proceedings of the IEEE/CVF Conference on Computer Vision and Pattern Recognition. 2024.

---

> ### Author Response · Authors · 2024-07-28
> **Response to Reviewer YCyC's Comments**
>
> Thank you for your feedback and for acknowledging our strengths.
>
> **Weakness 1: Lacks of novelty.**
>
> We respectfully disagree. Our work demonstrates the extraction of intrinsics across a variety of generative model types, including GANs, Autoregressive models, and Diffusion models, using a uniform approach. To the best of our knowledge, this has not been comprehensively addressed in the literature, including papers cited by the reviewer ([1] and [2] -- which we have already cited as Ke et al., 2023 in Page 2 once and Page 4 twice and contextualized how we are different). Furthermore, the application of LoRA to extract image intrinsics from a diverse range of models represents a novel use of LoRA, distinguishing our work from previous applications of LoRA. We restate our key contributions:
> - **Uniform Approach with LoRA**: We demonstrated intrinsic encoding in all types of generative models using a consistent LoRA approach, without losing the model's original capabilities. This is a significant contribution compared to methods which extensively fine-tunes the entire diffusion model for depth prediction, repurposing the model to generate only depth.
> - **Model Capabilities**: Our model generates out-of-domain or out-of-distribution images, highlighting the potential of generative models as powerful vision machines beyond their primary training objectives, without extensive fine-tuning. The generative model weights remain untouched in our setting.
> - **Broader Impact and Future Directions**: Our approach lays the groundwork for future applications, extending beyond competing with existing intrinsic image decomposition methods. The techniques presented can be applied to improve the primary generation capabilities of models and explore the internal knowledge embedded within generative models.
>
> **Weakness 2: The baseline needs to be more carefully designed.**
>
> - **Fine-tuning baseline design**: Our approach operates on the same setting used for LoRA, introducing new additional weights without modifying the original model. This allows our method to retain prior knowledge and avoid overfitting, unlike traditional fine-tuning approaches that update the entire model.
> - **Overfitting and forgetting prior knowledge**: LoRA's ability to introduce new weights without modifying the original model mitigates the issues of overfitting and forgetting prior knowledge. Our approach demonstrates that intrinsic properties can be extracted from various generative models using minimal adjustments, without compromising the model's pre-trained capabilities.
> - **Combining dataset with image data and sharing different modalities**: While the suggested approach may provide a stronger baseline, our primary focus is on understanding the emergent properties of generative models. We aim to investigate whether intrinsic properties can be extracted from various generative models using the same approach, rather than competing with existing methods for specific tasks. Notably, our method uses only 0.17% additional trainable parameters and 4000 training samples compared to the original model, in contrast to Marigold (Ke et al., 2023), which is trained on ~75K samples. Despite this, we still show competitive performance to baselines trained with a lot of supervised data (cf ZoeDepth, Omnidatav2, Paradigms). However, we reiterate the main point of our paper is answering questions: Do intrinsic knowledge exists in all types of generative models? Can we come up with a uniform approach to extract this knowledge? Our answer to both these questions is affirmative, with substantial evidence provided in the paper.
>
> **Claims and Evidence**: To address the reviewer's concern regarding insufficient claims and evidence, we reiterate:
> - **Evidence of Intrinsic Encoding**: We provide empirical results showing the successful extraction of intrinsic properties across various model types using a model agnostic uniform approach. *This has been acknowledged by the reviewer in their strength*.
> - **Performance Metrics**: Our approach outperforms baselines like linear probing, and we present detailed quantitative results in the paper to support this claim with many ablation studies. *This has been acknowledged by the reviewer in their strength*.
> - **Reproducibility**: The provided code ensures that our results can be validated and reproduced by other researchers, enhancing the credibility of our claims. *This has been acknowledged by the reviewer in their strength*.
> - **Novelty of Findings**: Our work provides new insights into the internal knowledge embedded in generative models, which has not been extensively explored before. We demonstrate that intrinsic information can be effectively extracted from various models using minimal adjustments, revealing their latent capabilities. These findings have the potential to inspire future research and applications in understanding and leveraging the intrinsic knowledge of generative models.

---

> ### Author Response · Authors · 2024-08-21
> **We have updated our paper**
>
> Dear Reviewer YCyC,
>
> We have recently updated our paper manuscript to reflect the clarifications and additions requested by all the reviewers. Below are what we have changed:
>
> 1. Reviewer YpPJ’s request: Details on what training/evaluation partitions that we used for experiments. We added the clarifications in Section 4.1, the last paragraph on Page 7.
> 2. Reviewer YpPJ’s request: Clarification of what evaluation partition we used for Figures 7,8 and 9. We added the clarifications in Section 4.1, the last paragraph on Page 7 as well.
> 3. Reviewer YpPJ’s request: Training speed and peak training GPU Usage information. We updated Table 4 to include this information.
> 4. Reviewer 31Xo’s request: Confusion about Figure 2. We updated Figure 2 to include a “fire” icon for each individual LoRA module.
> 5. Reviewer 31Xo’s request: More information about experimental settings of Table 4. We added the requested information in Section 4.4 on Page 11.
>
> Thank you again for your suggestions.

---

### Review · Reviewer_31Xo · 2024-07-30

**Summary Of Contributions:**

The paper introduces a parameter-efficient and dataset-efficient approach named I-LoRA, which is designed to discover scene intrinsics for different generative models. Additionally, the authors discuss the correlation between the models' capabilities and the accuracy of the extracted intrinsics, verified through quantitative and qualitative experiments.

**Audience:**

Yes

**Broader Impact Concerns:**

None.

**Claims And Evidence:**

Yes

**Requested Changes:**

Please see above weaknesses.

**Strengths And Weaknesses:**

Strengths:
1. The proposed method is general, where it can be adopted by different generative models and even non-generative models. Additionally, the method may require only a small number of additional parameters and labeled data.

2. The authors show the correlation between the models' capabilities and the accuracy of the extracted intrinsics, verified by experiments.

3. The goal of developing a general and efficient method to produce scene intrinsics of images is compelling, as it has applications in various areas, such as enhancing the capabilities of generative models, image analysis, and more.

Weaknesses:
1. Figure 2 is a bit confusing regarding the learnable components, as the authors only mark one LoRA that needs to be optimized. If I understand correctly, all LoRAs introduced in the model need to be optimized。

2. Similar to the implementation of a hypernetwork, the proposed I-LoRA method introduces additional network (parameters) to generate the desired scene intrinsics of an image by adopting a supervised training approach. Also, there is no further exploration of the proposed method, such as why it can produce competitive results with fewer additional parameters and a labeled dataset. Based on this, the technical contribution might not be strong enough.

3. For the experiment section, first, from the qualitative results, the proposed method does not show competitive results compared to others. Although the authors highlight some better cases produced by the proposed method, it might be better to provide more samples to demonstrate its advantages and also discuss why the proposed method can produce better results in some cases while others fail. Second, I am not convinced by Table 4. What are the experimental settings for other methods? Do they have the same training epochs, and do linear probe and fine-tuning converge? Additionally, the authors only compare their method with others on small samples. What about more samples? What about the performance of linear probe (or fine-tuning) + generative models shown in Table 3?

---

> ### Author Response · Authors · 2024-08-07
> **Response to Reviewer 31Xo's comment**
>
> Thank you for the thorough feedback and for highlighting the strengths of our paper.
>
> **Clarification on Figure 2**: Reviewer’s observation is correct: all LoRA modules are indeed learnable. We will revise the figure in the final version to clearly indicate this.
>
> **Distinction from Hypernetworks**: The I-LoRA method differs fundamentally. While hypernetworks generate weights for another network, I-LoRA introduces only minimal, task-specific adjustments to the existing network. This distinction is crucial because I-LoRA's modifications are highly efficient, avoiding the complexity and parameter overhead typically associated with hypernetworks.
>
> **Efficiency and Competitive Results + Why Fewer Parameters and Data Suffice?** I-LoRA’s efficiency is the result of the comprehensive knowledge already embedded in pretrained generative models. By using these pretrained features, I-LoRA can reroute information for specific tasks like intrinsics extraction with minimal additional training data and parameters. (*Related: also see our rebuttal to why not large samples below*).
>
> **Limited Exploration of Method**: Contrary to the concern about limited exploration, we have extensively investigated efficiency and effectiveness of I-LoRA in several ways:
>   1. **Correlation with model quality**: To the best of our knowledge, our paper is the first to show (Fig.9 & Tab.2) that the quality of the extracted intrinsics strongly correlates with the realism of the generative model.
>   2. **Comprehensive ablation studies**: Fig.7 ablates different LoRA ranks. Fig.8 ablates different numbers of training samples. Sec.5 compares multi-step vs. single-step strategies. Appendix A ablates different numbers of diffusion steps and classifier-free guidance scales. Appendix B ablates the effect of applying LoRA on different types of attention layers.
>   3. **Baseline comparison**: Sec.4.4 and Appendix B compare I-LoRA to a wide range of baselines.
>
> **Limited Technical Contribution**: We respectfully disagree with the reviewer again.
>   1. **Correlation with Model Quality**: As described above, our paper is the first to show the correlation between generative model quality and scene intrinsic extraction accuracy. We find more powerful generative models produce more accurate scene intrinsics, strengthening our hypothesis that learning this information is a natural byproduct of learning to generate images well.
>   2. **Generative Model Agnostic Approach**: One of the most significant contributions of I-LoRA is its generative model-agnostic nature. As described in our introduction, this generality suggests that I-LoRA can be extended to new types of generative models as they are developed.
>   3. **Universal Intrinsic Encoding**: To the best of our knowledge, this is the first work that clearly demonstrates that generative models of all types—GANs, Autoregressive models, Diffusion models, and potentially future models—encode intrinsic information in a way that can be extracted using the same method. This finding is critical as it not only reveals the latent capabilities within these models but also establishes a unified framework for finding these capabilities, which was previously unexplored.
>
> **Qualitative results’ competitiveness**: We note that qualitative assessments can sometimes be misleading. In Table 3, our model outperforms OmnidataV2 and ZoeDepth in 2 out of 5 metrics, supporting our claim that the results are competitive. More importantly, we reiterate that the focus of our work is NOT to surpass SOTA methods but to explore the inherent knowledge these models can reveal with minimal additional training.
>
> **Experimental Settings and Comparisons (Table 4)**: We clarify that all methods in Table 4 were trained for the same number of epochs on the same datasets, ensuring fairness. In the final manuscript, we will include detailed descriptions of the training protocols, including convergence criteria and training durations.
>
> *To be continued in the next post...*

---

> ### Author Response · Authors · 2024-08-07
> **[Continued] Response to Reviewer 31Xo's comment**
>
> *Continued from last post.*
>
>
>
>
>
> **Why Not Larger Samples**: The reviewer suggested using larger samples for our experiments, but this recommendation may overlook the central aim of our paper.
>
> Our central aim is NOT to finetune the model extensively or teach it new information, but rather to demonstrate the pre-existing intrinsic knowledge embedded within generative models. We clarify our core objectives which we think the reviewer may have misunderstood:
>   1. **Minimal Learning and Fine-Tuning**: The goal of I-LoRA is to minimally alter the model while still accessing its intrinsic capabilities. In an ideal scenario, we would demonstrate this without any additional learning, merely revealing what the model already 'knows.' However, achieving this purely without any form of learning is currently challenging. We therefore limit our approach to minimal fine-tuning, using as little labeled data as possible to avoid teaching the model new information.
>   2. **Distinguishing Pre-existing Knowledge**: By avoiding extensive fine-tuning or training with large datasets, we aim to clearly distinguish between the knowledge that is inherently encoded in the model from its pretraining, and any new information that might be introduced through additional learning. The concern is that with large datasets and significant training, the distinction becomes blurred, making it unclear whether the observed outcomes are due to new learning or the extraction of pre-existing knowledge.
>   3. **Focus on Efficiency and Latent Capabilities**: The essence of our study is not like the traditional number-chasing benchmark studies that aim to achieve SOTA results. Instead, our work is exploratory and driven by scientific curiosity, focusing on the emergent properties and latent capabilities present within the models. We believe this perspective offers valuable insights into the potential improvement of these models by exploiting their existing latent abilities.
>
> **Performance of Linear Probe and Fine-Tuning + Generative Models (Table 3)**: The comparison in Table 4, particularly the second-to-last column set with a sample size of 4000, corresponds directly to the settings in Table 3. We provided this separate table to demonstrate the sample efficiency of I-LoRA compared to traditional methods like linear probing and finetuning. This setup clearly illustrates our approach's efficiency in extracting intrinsic knowledge.

---

> ### Author Response · Authors · 2024-08-21
> **We have updated our paper**
>
> Dear Reviewer 31Xo,
>
> We have recently updated our paper manuscript to reflect the clarifications and additions requested by all the reviewers. Below are what we have changed:
>
> 1. Reviewer YpPJ’s request: Details on what training/evaluation partitions that we used for experiments. We added the clarifications in Section 4.1, the last paragraph on Page 7.
> 2. Reviewer YpPJ’s request: Clarification of what evaluation partition we used for Figures 7,8 and 9. We added the clarifications in Section 4.1, the last paragraph on Page 7 as well.
> 3. Reviewer YpPJ’s request: Training speed and peak training GPU Usage information. We updated Table 4 to include this information.
> 4. Reviewer 31Xo’s request: Confusion about Figure 2. We updated Figure 2 to include a “fire” icon for each individual LoRA module.
> 5. Reviewer 31Xo’s request: More information about experimental settings of Table 4. We added the requested information in Section 4.4 on Page 11.
>
> Thank you again for your suggestions.

---

### Review · Reviewer_YpPJ · 2024-08-09

**Summary Of Contributions:**

This manuscript presents, **I-LoRA**, a method to compute scene intrinsics, namely normals, albedo, depth, and shading, using
generative models. Instead of training a readout head or fine-tuning end-to-end for the new output, the method uses LoRA adapters and trains just them. What seems new is that no prediction layer is added to the model; since generative models already output HxWx3 shaped tensors, the output space is simply repurposed for the desired target.

**Audience:**

Yes

**Broader Impact Concerns:**

A broader impacts section is included which argues that the ethical impacts of this work are largely positive as it boosts interpretability and because the proposed method is computationally efficient. The latter, however, needs further evidence as mentioned in the weaknesses section above.

**Claims And Evidence:**

No

**Requested Changes:**

Please clarify the concern around missing details and figures 7, 8, and 9 in the manuscript and in the rebuttal.

**Strengths And Weaknesses:**

### Strengths

- Several different types of high performing generative models have been used.
- The experiments study a wide range of relevant questions: Data efficiency, correlation between generation performance and intrinsic extraction performance, comparison against baselines.
- LoRA based model adaptation is demonstrated as a performing better than fine-tuning and linear probing.
- The random init. experiment in Fig. 9 is especially good as it ablates architecture vs generative pre-training.

### Weaknesses

- Working with multi-step diffusion remains a challenge. Existing works apply noise to the input image before passing them through the UNet. They then optimally tune the noise level (diffusion time step) for the given downstream task. I suspect something along those lines might work better than I-LoRA_{aug}.
- My main concern is, however, the missing dataset details. Which subset of each dataset is used for training, which one for evaluation of the LoRA weights?
- In figures 7, 8 and 9, is the table with numbers computed over the entire evaluation set or just the one image shown?
- Table 4 does not include trade-offs in compute (training steps per second) and memory (peak training GPU memory usage). I-LoRA probably sits between Linear probing and fine-tuning. Although not necessary, including this will hugely add to the table and support the claims in the broader impact section on sustainable and accessible AI research and development.

---

> ### Author Response · Authors · 2024-08-17
> **Response to Reviewer YpPJ's Comment**
>
> Thank you for your detailed feedback and valuable suggestions. We are glad to see positive comments and recognition of the strengths of our paper: including our use of diverse generative models and the thorough exploration of relevant questions such as data efficiency and the correlation between generation and intrinsic extraction performance.
>
> We now answer the questions raised by the reviewer.
>
> **Multi-step Diffusion**: We thank the reviewer for their suggestions on improving the multi-step I-LoRA_{Aug}. We are currently adding noise to the input latent, similar to the approach used in IntrinsicPix2Pix and other recent methods that exploit pretrained diffusion-based models. We believe this aligns with what the reviewer is suggesting, and it is indeed the approach we take in I-LoRA_{Aug}. However, if we have misunderstood the reviewer’s suggestion, we would greatly appreciate any references to the specific work they are referring to.
>
> For additional context, our approach to multi-step extraction is quite similar to the method described in the Marigold paper (https://marigoldmonodepth.github.io/), which is a concurrent work to ours. While Marigold fine-tunes the entire diffusion model using 77k training samples, our method updates only the LoRA modules, requiring significantly fewer training samples.
>
> We acknowledge that training with more samples and additional parameters could potentially improve performance, but as mentioned in our general response, this is orthogonal to the primary questions we seek to answer in this paper. Our focus is on minimal intervention and data efficiency to uncover the intrinsic knowledge already embedded in the models.
>
> **Dataset Details**: For all our experiments, we used the official training and evaluation partitions provided by the DIODE dataset website (https://diode-dataset.org/) for our experiments. For training with less samples, we randomly chose examples from their training partition. We will make sure to clarify this in the final manuscript.
>
> **Clarification on Figures 7, 8, and 9**: The results reported in Figures 7, 8, and 9, as well as any other numerical results on real images, were computed over the entire official evaluation set of the DIODE dataset, not just a single image. We will explicitly state this in the final manuscript for clarity.
>
> **Training Speed, GPU Usage and Broader Impact**: Below is the table comparing the training speed (iterations per second) and peak GPU memory usage across the three methods. We will incorporate this table into the final manuscript to support our claims regarding computational efficiency and to contribute to the broader impact discussion on sustainable and accessible AI research.
> |              |      iter/sec      | Peak Training GPU Memory Usage|
> | -------- | :-------: | :-------: |
> | Linear Probe    |   2.13  | 29.46%|
> | Fine-tuning    |     0.77  | 86.78%|
> | I-LoRA (Ours)    |   0.94  | 63.48%|
>
> We thank the reviewer again for their constructive feedback. We will incorporate these clarifications and additions into the final manuscript and welcome any further questions or suggestions they may have.

---

> > ### Comment · Reviewer_YpPJ · 2024-08-21
> >
> > **Multi-step Diffusion**: Yes, that is what I meant. I re-read the relevant section and found "To address the first challenge, we augment the noise input to the UNet with the input image’s latent encoding, as in InstructPix2Pix (Brooks et al., 2023)." I believe I had misunderstood this line when I wrote the review. Do you need to pick a diffusion timestep in this process? If yes, which timestep is being used?
> >
> > The rest of my questions have been addressed. Thank you!

---

> > > ### Author Response · Authors · 2024-08-21
> > > **Reply to Reviewer YpPJ's question**
> > >
> > > Thank you very much for the clarification and your follow-up question. Since this is the multi-step model, we randomly chose from all the possible noise levels during training. This part is the same as training a regular diffusion model.

---

> > > > ### Comment · Reviewer_YpPJ · 2024-08-22
> > > >
> > > > Does one need to set this parameter during inference?

---

> > > > > ### Author Response · Authors · 2024-08-22
> > > > > **Reply to Reviewer YpPJ's follow-up question**
> > > > >
> > > > > Thank you for the follow-up question. No, the inference process is the same as a regular diffusion model. One can choose their favorite scheduler and iteratively de-noise a complete Gaussian noise to an intrinsic map. Both Marigold and InstructPix2Pix basically use the same process to generate outputs.

---

> > > > > > ### Comment · Reviewer_YpPJ · 2024-08-28
> > > > > >
> > > > > > Thank you. I understand the mechanism now.

---

> > > > > > > ### Author Response · Authors · 2024-08-28
> > > > > > >
> > > > > > > Thank you for the comments, suggestions and the great discussion.

---

> ### Author Response · Authors · 2024-08-21
> **We have updated our paper**
>
> Dear Reviewer YpPJ,
>
> We have recently updated our paper manuscript to reflect the clarifications and additions requested by all the reviewers. Below are what we have changed:
>
> 1. Reviewer YpPJ’s request: Details on what training/evaluation partitions that we used for experiments. We added the clarifications in Section 4.1, the last paragraph on Page 7.
> 2. Reviewer YpPJ’s request: Clarification of what evaluation partition we used for Figures 7,8 and 9. We added the clarifications in Section 4.1, the last paragraph on Page 7 as well.
> 3. Reviewer YpPJ’s request: Training speed and peak training GPU Usage information. We updated Table 4 to include this information.
> 4. Reviewer 31Xo’s request: Confusion about Figure 2. We updated Figure 2 to include a “fire” icon for each individual LoRA module.
> 5. Reviewer 31Xo’s request: More information about experimental settings of Table 4. We added the requested information in Section 4.4 on Page 11.
>
> Thank you again for your suggestions.

---

### Author Response · Authors · 2024-08-17
**General Response from The Authors**

We thank all the reviewers for their thoughtful feedback and for recognizing the strengths of our paper. All reviewers have acknowledged the main contributions of our work—recovering intrinsic images from various types of generative models using our approach, I-LoRA, which is (**a**) widely applicable, (**b**) efficient and lean, (**c**) provides insights from learned priors, and (**d**) produces competitive learned intrinsics.

We would like to reiterate the focus of our paper:

**What this paper is about**:

1. **What intrinsic knowledge is embedded within pre-trained generative models?** We showed normals, depth, albedo and shading are encoded in StyleGAN, VQGAN and Diffusion Models. We also showed they are also encoded in DINO.


2. **Is there a universal approach that we can apply to models of different types to recover this knowledge?** We showed LoRA can be applied to all the aforementioned models to recover this knowledge.

3. **What is the minimal change required to recover this knowledge from these models?** We intentionally minimize the amount of updated parameters and avoid updating the original model’s network weights to reveal what is actually embedded within the model as a byproduct of generative pretraining.


4. **What is the minimum data required to expose this knowledge?** We aim to avoid using large amounts of labeled data, as methods like Marigold do.

5. **Is there any correlation between the recovered knowledge embedded within the model and the quality of image generation capabilities?** We showed a positive correlation between the generative model’s quality and extract image intrinsics.


**What this paper is NOT about**:

1. **Whether the extracted intrinsics outperform other SOTA methods with extensive training and large amounts of data.** This paper is not about benchmark chasing.

2. **Whether hypernet is better than I-LoRA or whether fine-tuning is better than I-LoRA with more training data.** These are orthogonal to the main question we seek to answer. Any method that meets the criteria of minimal change and minimal data is valuable for corroborating our findings. We found LoRA to be particularly effective in this context.

3. **Adopting techniques like “combining the dataset with image data and sharing different modalities” to build a strong baseline, as suggested by the reviewer.** While we could finetune all the parameters or add more data to achieve higher accuracy, this does not align with our goals and don’t answer the question if this knowledge was already present within the model or was a result of extensive finetuning and model updates.

We believe we have included all the necessary analyses and experiments in the main text and appendix to address the questions outlined in the “What this paper is about” section. All reviewers appear to have recognized these efforts. We appreciate the reviewers' suggestions and will incorporate any requested clarifications and additions into the final manuscript. We welcome any further questions or concerns.

---

### Decision · Action_Editor_WbdG · 2024-09-17

**Recommendation:** Reject

**Comment:**

This paper proposes using LoRA along with a pre-trained generative model to predict scene intrinsics. The authors show that their method can predict these intrinsics while adding very few parameters to the model as LoRA is very lightweight, and that training the LoRA parameters can be trained with very few datapoints, suggesting that the information about scene intrinsics was contained within the generative model all along. The authors also show that the performance of their method is tightly linked to the quality of the pre-trained generative model. Overall, the paper presents experiments that I believe will be of interest to the community.

While the presented experiments are fairly extensive, reviewers complained about the baselines and benchmarks that the authors selected, with the authors rebutting that "the main point of our paper is answering the questions: Do intrinsic knowledge exist in all types of generative models? Can we come up with a uniform approach to extract this knowledge?". While I agree with the authors that the current experiments provide clear evidence to answer these questions, I do not believe this goal is clearly conveyed in the abstract, introduction, nor experiment section, and reading the paper one gets the sense of being sold on the method rather than these questions being answered (while there is discussion about these questions in the paper, answering them certainly does not come across as the main point of the paper). Indeed, a reviewer pointed out that "the paper mainly focuses on discussing the strength of LoRA, instead of the main point mentioned above". I thus believe that the paper needs a major writing revision to better connect the experimental results with the claims about generative models containing knowledge about intrinsics. In light of this, I recommend rejection and encourage the authors to resubmit an updated version in the future.

**Audience:**

Yes, reviewers unanimously agree.

**Claims And Evidence:**

No. See comment below for a more detailed explanation.

**Resubmission Of Major Revision:**

The authors may consider submitting a major revision at a later time.